# Nuclear Norm Regularization for Deep Learning

**Christopher Scarvelis**
MIT CSAIL
scarv@mit.edu

**Justin Solomon**
MIT CSAIL
jsolomon@mit.edu

## Abstract

Penalizing the nuclear norm of a function's Jacobian encourages it to locally behave like a low-rank linear map. Such functions vary locally along only a handful of directions, making the Jacobian nuclear norm a natural regularizer for machine learning problems. However, this regularizer is intractable for high-dimensional problems, as it requires computing a large Jacobian matrix and taking its SVD. We show how to efficiently penalize the Jacobian nuclear norm using techniques tailor-made for deep learning. We prove that for functions parametrized as compositions $f = g \circ h$, one may equivalently penalize the average squared Frobenius norms of $Jg$ and $Jh$. We then propose a denoising-style approximation that avoids Jacobian computations altogether. Our method is simple, efficient, and accurate, enabling Jacobian nuclear norm regularization to scale to high-dimensional deep learning problems. We complement our theory with an empirical study of our regularizer's performance and investigate applications to denoising and representation learning.

## 1 Introduction

Building models that adapt to the structure of their data is a key challenge in machine learning. As real-world data typically concentrates on low-dimensional manifolds, a good model $f$ should only be sensitive to changes to its inputs along the data manifold. One may encourage this behavior by regularizing $f$ so that its *Jacobian $Jf[x]$* has low rank. This causes $f$ to locally behave like a low-rank linear map and therefore be locally constant in directions that are in the kernel of $Jf[x]$.

How should we regularize $f$ so that its Jacobians have low rank? Directly penalizing $\text{rank}(Jf[x])$ during training is challenging, as the rank function is not differentiable. In light of this, the *nuclear norm* $\|Jf[x]\|_*$ is an appealing alternative: Being the $L^1$ norm of a matrix's singular values, it is the tightest convex relaxation of the rank function, and as it is differentiable almost everywhere, it can be included in standard deep learning pipelines.

One critical flaw in this strategy is its computational cost. The nuclear norm of a matrix is the sum of its *singular values*, so to penalize $\|Jf[x]\|_*$ in training, one must (1) compute the Jacobian of a typically high-dimensional map $f : \mathbb{R}^n \to \mathbb{R}^m$, (2) take the singular value decomposition (SVD) of this $m \times n$ matrix, (3) sum its singular values, and (4) differentiate through each of these operations. The combined cost of these operations is prohibitive for high-dimensional data. Consequently, nuclear norm regularization has yet to be widely adopted by the deep learning community.

This work shows how to efficiently penalize the Jacobian nuclear norm $\|Jf[x]\|_*$ using techniques tailor-made for deep learning. We first show that parametrizing $f$ as a composition $f = g \circ h$ – a feature common to *all deep learning pipelines* – allows one to replace the expensive nuclear norm $\|Jf[x]\|_*$ with two squared *Frobenius* norms $\|Jh[x]\|_F^2$ and $\|Jg[h(x)]\|_F^2$, which admit an elementwise computation that avoids a costly SVD. We prove that the resulting problem is *exactly* equivalent to the original problem with the nuclear norm penalty. We in turn approximate this by a denoising-style objective that avoids the Jacobian computation altogether. Our method is

38th Conference on Neural Information Processing Systems (NeurIPS 2024).

simple, efficient, and accurate – both in theory and in practice – and enables Jacobian nuclear norm regularization to scale to high-dimensional deep learning problems.

We complement our theoretical results with an empirical study of our regularizer's performance on synthetic data. As the Jacobian nuclear norm has seldom been used as a regularizer in deep learning, we propose applications of our method to unsupervised denoising, where one trains a denoiser given a dataset of noisy images without access to their clean counterparts, and to representation learning. **Our work makes the Jacobian nuclear norm a feasible component of deep learning pipelines**, enabling users to learn locally low-rank functions unencumbered by the heavy cost of naïve Jacobian nuclear norm regularization.

## 2  Related work

**Nuclear norm regularization.**   As penalizing the nuclear norm in matrix learning problems encourages low-rank solutions, nuclear norm regularization (NNR) has been widely used throughout machine learning. Rennie and Srebro [2005] propose NNR for collaborative filtering, where one attempts to predict user interests by aggregating incomplete information from a large pool of users. Candès et al. [2011] introduce robust PCA, which decomposes a noisy matrix into low-rank and sparse parts and uses nuclear norm regularization to learn the low-rank part. Cabral et al. [2013], Dai et al. [2014] use NNR to regularize the ill-posed structure-from-motion problem, which recovers a 3D scene from a set of 2D images.

A line of work beginning with Candès and Recht [2012] takes inspiration from compressed sensing and studies the conditions under which a low-rank matrix can be perfectly recovered from a small sample of its entries via nuclear norm minimization. This work was followed by Candès and Tao [2010], Keshavan et al. [2009], Recht [2011], which progressively sharpen the bounds on the number of samples required for exact recovery. In parallel, Cai et al. [2010] propose singular value thresholding (SVT), a simple algorithm for approximate nuclear norm minimization that avoids solving a costly semidefinite program as with earlier algorithms. However, SVT still requires computing a singular value decomposition in each iteration, which is an onerous requirement for large matrices. Motivated by this challenge, Rennie and Srebro [2005] show how to convert a nuclear norm-regularized matrix learning problem into an equivalent non-convex problem involving only squared Frobenius norms. Our work generalizes their method to non-linear learning problems.

**Jacobian regularization in deep learning.**   Sokolić et al. [2017], Varga et al. [2017], Hoffman et al. [2020] penalize the spectral and Frobenius norms of a neural net's Jacobian with respect to its inputs to improve classifier generalization, particularly in the low-data regime. Similarly, Jakubovitz and Giryes [2018] fine-tune neural classifiers with Jacobian Frobenius norm regularization to improve adversarial robustness. Unlike our work, these papers do not consider nuclear norm regularization.

Neural ODEs [Chen et al., 2018] parametrize functions as solutions to initial value problems with neural velocity fields. Ensuring that the learned dynamics are well-conditioned minimizes the number of steps required to accurately solve these ODEs. To this end, Finlay et al. [2020] penalize the squared Frobenius norm of the velocity field's Jacobian and observe a tight relationship between the value of this norm and the step size of an adaptive ODE solver. Kelly et al. [2020] extend this approach by regularizing higher-order derivatives as well.

Finally, it has been observed as early as Webb [1994], Bishop [1995] that training a neural net on noisy data approximately penalizes the squared Frobenius norm of the network Jacobian at training data. Inspired by this observation, Vincent et al. [2008, 2010] propose denoising autoencoders for representation learning, and Rifai et al. [2011] propose directly penalizing the squared Frobenius norm of the encoder Jacobian to encourage robust latent representations. Alain and Bengio [2014] show that an autoencoder trained with a penalty on the squared Frobenius norm of its Jacobian learns the score of the data distribution for small regularization values. Recently, Kadkhodaie et al. [2024] employ a similar analysis of the denoising objective to study the generalization of diffusion models.

**Denoising via singular value shrinkage.**   While a full survey of the denoising literature is out of scope (see e.g. Elad et al. [2023]), we highlight a handful of works that employ *singular value shrinkage* (SVS) to denoise low-rank data corrupted by isotropic noise given a noisy data matrix. SVS denoises a noisy data matrix $Y$ by applying a shrinkage function $\phi$ to its singular values $\sigma_d$.

This function *shrinks* small singular values of $Y$ while leaving larger singular values untouched. Shabalin and Nobel [2010] study the conditions under which it suffices to rescale the noisy data's singular values while preserving its singular vectors. Gavish and Donoho [2014] study the optimal hard thresholding policy, which sets all singular values below a threshold to 0 while preserving the rest. Nadakuditi [2014] considers general shrinkage policies whose error is measured by the squared Frobenius distance between the true and denoised data matrix. Gavish and Donoho [2017] then study optimal shrinkage policies under a larger class of error measures. All of these methods assume that the clean data matrix is low-rank and hence that the data globally lies in a low-dimensional subspace. Under the manifold hypothesis, real-world data such as images typically lie on low-dimensional manifolds and hence *locally* lie in low-dimensional subspaces. Inspired by this observation, we use our method in Section 5 to learn an image denoiser given a dataset of exclusively noisy images.

## 3 Method

In this section, we introduce our method for Jacobian nuclear norm regularization. We first prove that for functions parametrized as compositions $f = g \circ h$ (which include *all non-trivial neural networks*), one may replace the expensive nuclear norm penalty $\|Jf[x]\|_*$ in a learning problem with the average of two squared *Frobenius* norms $\frac{1}{2}\left(\|Jh[x]\|_F^2 + \|Jg[h(x)]\|_F^2\right)$ and obtain an equivalent learning problem. These squared Frobenius norms admit an elementwise computation that avoids a costly SVD in computing $\|Jf[x]\|_*$. As the Jacobian computation is itself costly for large neural nets, we use a first-order Taylor expansion and Hutchinson's trace estimator [Hutchinson, 1989] to estimate the Frobenius norm terms. Computing a loss with our regularizer costs as few as two additional evaluations of $g$ and $h$ per sample, allowing our method to scale to large neural networks with high-dimensional inputs and outputs.

### 3.1 Preliminaries

Any matrix $A \in \mathbb{R}^{m \times n}$ admits a *singular value decomposition* (SVD) $A = U\Sigma V^\top$, where $U \in \mathbb{R}^{m \times m}$ and $V \in \mathbb{R}^{n \times n}$ are orthogonal matrices, and $\Sigma \in \mathbb{R}^{m \times n}$ is a diagonal matrix storing the *singular values* $\sigma_i \geq 0$ of $A$. The rank of $A$ is equal to its number of non-zero singular values.

The *nuclear norm* $\|A\|_*$ of a matrix $A \in \mathbb{R}^{m \times n}$ is the sum of its singular values: $\|A\|_* = \sum_i \sigma_i$. As $\sigma_i \geq 0$ by definition, $\|A\|_*$ is also the $L^1$ norm of its vector of singular values. Just as $L^1$ regularization steers learning problems towards solutions with many zero entries, nuclear norm regularization steers *matrix* learning problems towards *low-rank* solutions with many zero singular values. For this reason, nuclear norm regularization has seen widespread use in problems ranging from collaborative filtering [Rennie and Srebro, 2005, Candès and Recht, 2012] to robust PCA [Candès et al., 2011] to structure-from-motion [Cabral et al., 2013, Dai et al., 2014].

However, $\|A\|_*$ is costly to compute because it requires a cubic-time SVD. Furthermore, even efficient algorithms for nuclear norm minimization such as Cai et al. [2010]'s *singular value thresholding* require one SVD per iteration, which is prohibitive for very large matrices $A$. Motivated by this computational challenge, Rennie and Srebro [2005, Lemma 1] state that one can compute $\|A\|_*$ by solving a non-convex optimization problem:

$$\|A\|_* = \min_{U,V:UV^\top=A} \frac{1}{2}\left(\|U\|_F^2 + \|V\|_F^2\right). \tag{1}$$

For completeness, we prove this identity in Appendix A.1. Using this result, Rennie and Srebro [2005] show that the following problems are equivalent:

$$\min_{A \in \mathbb{R}^{m \times n}} \ell(A) + \eta\|A\|_* = \min_{\substack{U \in \mathbb{R}^{m \times r} \\ V \in \mathbb{R}^{n \times r}}} \ell(UV^\top) + \frac{\eta}{2}\left(\|U\|_F^2 + \|V\|_F^2\right), \tag{2}$$

where $r = \min(m, n)$ and $\ell$ is a generic differentiable loss function. As $\frac{1}{2}\|U\|_F^2$ and its derivative $\nabla_U \frac{1}{2}\|U\|_F^2 = U$ can be computed elementwise without a costly SVD, the RHS objective in (2) is amenable to gradient-based optimization.

## 3.2 Our key result

Equation (2) enables the use of simple and efficient gradient-based methods for learning a low-rank linear map by parametrizing the matrix $A$ as a *composition* $A = UV^\top$ of linear maps $U$ and $V^\top$. In deep learning, one is interested in learning *non-linear* functions that are parametrized by compositions of simpler functions. Such functions $f$ are differentiable almost everywhere, so they are *locally* well-approximated by linear maps specified by their *Jacobians* $Jf[x]$.

Encouraging the learned function to have a low-rank Jacobian is a natural prior: It corresponds to learning a function that locally behaves like a low-rank linear map. Such functions vary locally along only a handful of directions and are constant in the remaining directions. When the training data is supported on a low-dimensional manifold, these directions correspond to the tangents and normals, respectively, to the data manifold. One may implement this low-rank prior on $Jf$ by solving the following optimization problem:

$$\inf_{f:\mathbb{R}^n \to \mathbb{R}^m} \mathbb{E}_{x \sim \mathcal{D}(\Omega)} \left[ \ell(f(x), x) + \eta \|Jf[x]\|_* \right], \tag{3}$$

where $\ell$ is a generic differentiable loss function and $\mathcal{D}(\Omega)$ is a data distribution supported on $\Omega \subseteq \mathbb{R}^n$. If $f$ is parametrized as a neural network and $n, m$ are large, this problem is costly to optimize via stochastic gradient descent, as $Jf[x] \in \mathbb{R}^{m \times n}$ and computing the subgradient of $\|Jf[x]\|_*$ requires a cubic-time SVD. In fact, simply *storing* $Jf[x]$ in memory is often intractable for large $n, m$, which is typical when $f$ is an image-to-image map. For example, if $f$ is a denoiser operating on $1024 \times 1024$ RGB images, its inputs are $3 \times 1024 \times 1024 = 3{,}145{,}728$-dimensional, and $Jf[x]$ occupies nearly 40 TB of memory.

To address these challenges, we first prove a theorem generalizing (2) to non-linear functions. We then show how to avoid computing $Jf[x]$ altogether using a first-order Taylor expansion and Hutchinson's estimator. **Our primary contribution is the following result:**

**Theorem 3.1** *Let $D(\Omega)$ be a data distribution supported on a compact set $\Omega \subseteq \mathbb{R}^n$ with measure $\mu$ that is absolutely continuous with respect to the Lebesgue measure on $\Omega$. Let $\ell \in C^1(\mathbb{R}^m \times \mathbb{R}^n)$ be a continuously differentiable loss function. Then,*

$$\inf_{f \in C^\infty(\Omega)} \mathbb{E}_{x \sim \mathcal{D}(\Omega)} \left[ \ell(f(x), x) + \eta \|Jf[x]\|_* \right]$$
$$= \inf_{\substack{h \in C^\infty(\Omega) \\ g \in C^\infty(h(\Omega))}} \mathbb{E}_{x \sim \mathcal{D}(\Omega)} \left[ \ell(g(h(x)), x) + \frac{\eta}{2} \left( \|Jg[h(x)]\|_F^2 + \|Jh[x]\|_F^2 \right) \right]. \tag{4}$$

On the left-hand side, we learn a function $f : \mathbb{R}^n \to \mathbb{R}^m$ given fixed input and output dimensions $n, m$. On the right-hand side, we learn functions $h : \mathbb{R}^n \to \mathbb{R}^d$ and $g : \mathbb{R}^d \to \mathbb{R}^m$ with $n, m$ fixed but optimize over the inner dimension $d$. We prove this theorem in Appendix A.2 and sketch the proof below. Theorem 3.1 shows that by parametrizing $f$ as a composition of $g$ and $h$ – a feature common to all deep learning pipelines – one may learn a *locally* low-rank function without computing expensive SVDs during training.

**Proof sketch.** We denote the left-hand side objective by $E_L(f)$ and its inf by $(L)$; we denote the right-hand side objective by $E_R(g, h)$ and its inf by $(R)$. We prove that $(L) \leq (R)$ and $(R) \leq (L)$.

$(L) \leq (R)$ is the easy direction. The basic observation is that if $f = g \circ h$, then $Jf[x] = Jg[h(x)]Jh[x]$ by the chain rule. Equation (1) then implies that

$$\|Jf[x]\|_* = \min_{U, V : UV^\top = Jf[x]} \frac{1}{2} \left( \|U\|_F^2 + \|V\|_F^2 \right) \leq \frac{1}{2} \left( \|Jg[h(x)]\|_F^2 + \|Jh[x]\|_F^2 \right).$$

$(R) \leq (L)$ is the hard direction. The proof strategy is as follows:

1. We begin with a function $f_m \in C^\infty(\Omega)$ such that $E_L(f_m)$ is arbitrarily close to its inf over $C^\infty(\Omega)$. We use $f_m$ to construct parametric families of affine functions $g_m^z, h_m^z$ whose composition is a good local approximation to $f_m$ in a neighborhood of $z \in \Omega$, both pointwise and in terms of the contributions to $E_R(g_m^z, h_m^z)$ and $E_L(f_m)$, resp., due to $x \in \Omega$ near $z$.

2. We then stitch together these local approximations to form a sequence of global approximations $g_m^k, h_m^k$ to $f_m$. These functions are piecewise affine and hence not regular enough to lie in $C^\infty(\Omega)$ as required by the right-hand side of Equation (4).

3. Finally, mollifying the piecewise affine functions $g_m^k, h_m^k$ yields a minimizing sequence of $C^\infty(\Omega)$ functions $g_{m,\epsilon}^k, h_{m,\epsilon}^k$ such that $E_R(g_{m,\epsilon}^k, h_{m,\epsilon}^k)$ approaches the inf of $E_L$.

While Theorem 3.1 shows how to regularize a learning problem with the Jacobian nuclear norm without a cubic-time SVD, the Jacobian computation incurs a quadratic time and memory cost, which remains heavy for high-dimensional learning problems. To mitigate this issue, the following section shows how to approximate the $\|Jg[h(x)]\|_F^2$ and $\|Jh[x]\|_F^2$ terms in (4).

### 3.3 Estimating the Jacobian Frobenius norm

When $f : \mathbb{R}^n \to \mathbb{R}^m$ is a function between high-dimensional spaces, $Jf[x]$ is an $m \times n$ matrix that is costly to compute and to store in memory. Previous works employing Jacobian regularization for neural networks have noted this issue and proposed stochastic approximations based on Jacobian-vector products (JVP) against random vectors [Varga et al., 2017, Hoffman et al., 2020]. As JVPs may be costly to compute for large neural nets, we propose an alternative stochastic estimator that requires only evaluations of $f$ and analyze its error:

**Theorem 3.2** *Let $f : \mathbb{R}^n \to \mathbb{R}^m$ be continuously differentiable. Then,*

$$\sigma^2 \|Jf[x]\|_F^2 = \underset{\epsilon \sim \mathcal{N}(0,\sigma^2 I)}{\mathbb{E}} \left[ \|f(x+\epsilon) - f(x)\|_2^2 \right] + O(\sigma^2). \tag{5}$$

Similar results appear in the ML literature as early as Webb [1994]. Our proof in Appendix A.3 relies on a first-order Taylor expansion and Hutchinson's trace estimator; a similar proof is given by Alain and Bengio [2014]. In practice, we obtain accurate approximations to $\|Jf[x]\|_F^2$ by using a small noise variance $\sigma^2$ and rescaling the expectation in (5) by $\frac{1}{\sigma^2}$ to compensate. In Section 4, we also show that a single noise sample $\epsilon \sim \mathcal{N}(0, \sigma^2 I)$ suffices in practice.

Using this efficient approximation, we obtain the following regularizer:

$$\mathcal{R}(x; f) = \frac{1}{2\sigma^2} \underset{\epsilon \sim \mathcal{N}(0,\sigma^2 I)}{\mathbb{E}} \left[ \|g(h(x)+\epsilon) - g(h(x))\|_2^2 + \|h(x+\epsilon) - h(x)\|_2^2 \right], \tag{6}$$

where $f = g \circ h$. In practice, one may use a single draw of $\epsilon \sim \mathcal{N}(0, \sigma^2 I)$ per training iteration while maintaining good performance on downstream tasks; see e.g. the results in Section 5. In this case, our regularizer $\mathcal{R}(x; f)$ costs merely two additional function evaluations, enabling it to scale to large neural networks acting on high-dimensional data. In Section 4, we show that parametrizing $f_\theta$ as a neural net and solving

$$\inf_{f_\theta : \mathbb{R}^n \to \mathbb{R}^m} \underset{x \sim \mathcal{D}(\Omega)}{\mathbb{E}} \left[ \ell(f_\theta(x), x) + \eta \mathcal{R}(x; f_\theta) \right] \tag{7}$$

yields good approximations to the solution to (3) for problems where exact solutions are known. We then propose two applications of Jacobian nuclear norm regularization in Section 5.

## 4 Validation

In this section, we empirically validate our method on a special case of (3) for which closed-form solutions are known. We consider the following problem:

$$\inf_{f : \mathbb{R}^n \to \mathbb{R}} \int_{\mathbb{R}^n} \left[ \frac{1}{2} \|f(x) - \tau(x)\|_2^2 + \eta \|Jf[x]\|_* \right] dx, \tag{8}$$

where $\tau : \mathbb{R}^n \to \mathbb{R}$ is the indicator function of the unit ball in $\mathbb{R}^n$. As $f$ is a scalar-valued function in this problem, $Jf[x]$ is a *vector*, and $\|Jf[x]\|_* = \|\nabla f(x)\|_2$. This is an instance of the celebrated

*Rudin-Osher-Fatemi* (ROF) model for image denoising [Rudin et al., 1992]. Meyer [2001, p. 36] shows that the exact solution to (8) given this target function $\tau(x)$ is $f(x) := (1 - n\eta)\tau(x)$. This is a rescaled indicator function of the unit ball.

We parametrize $f$ as a multilayer perceptron (MLP) $f_\theta$ and solve (8) along with the problem using our regularizer:

$$\inf_{f_\theta:\mathbb{R}^n \to \mathbb{R}} \mathbb{E}_{x \sim \mathcal{D}(\Omega)} \left[ \frac{1}{2} \| f_\theta(x) - \tau(x) \|_2^2 + \eta \mathcal{R}(x; f_\theta) \right], \tag{9}$$

where $f_\theta = g_\theta \circ h_\theta$. We approximate the integral over $\mathbb{R}^n$ by Monte Carlo integration over a box $\Omega$ centered at the origin. We experiment with $n = 2$ and $n = 5$ and in each case depict results for a small and a large regularization value $\eta$. We track the objective values of problems (8) and (9) and show that they converge to the same value, as predicted by Theorem 3.1. We also track the absolute error of both problem's solutions across training iterations and plot solutions to each problem at convergence for the 2D case. We give full implementation details in Appendix B.1.

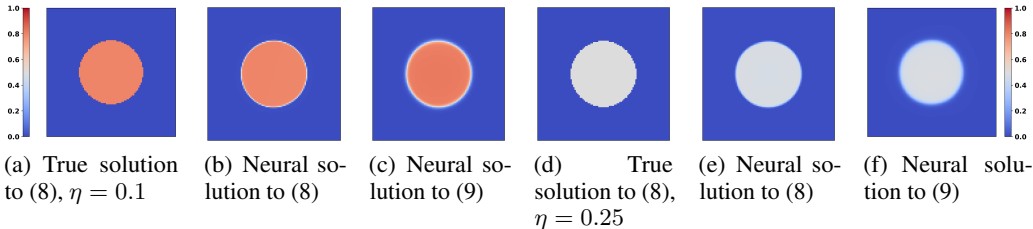

(a) True solution to (8), $\eta = 0.1$    (b) Neural solution to (8)    (c) Neural solution to (9)    (d) True solution to (8), $\eta = 0.25$    (e) Neural solution to (8)    (f) Neural solution to (9)

Figure 1: Comparison of exact and neural solutions to Problems (8) and (9) with $n = 2$ and $\eta = 0.1$ (first three plots) and $\eta = 0.25$ (last three plots). The $x-$ and $y-$ axes represent the inputs to $f_\theta$, and colors denote function values. Solving (9) recovers an accurate approximation to the true solution for both values of $\eta$ while requiring no Jacobian nuclear norm computations.

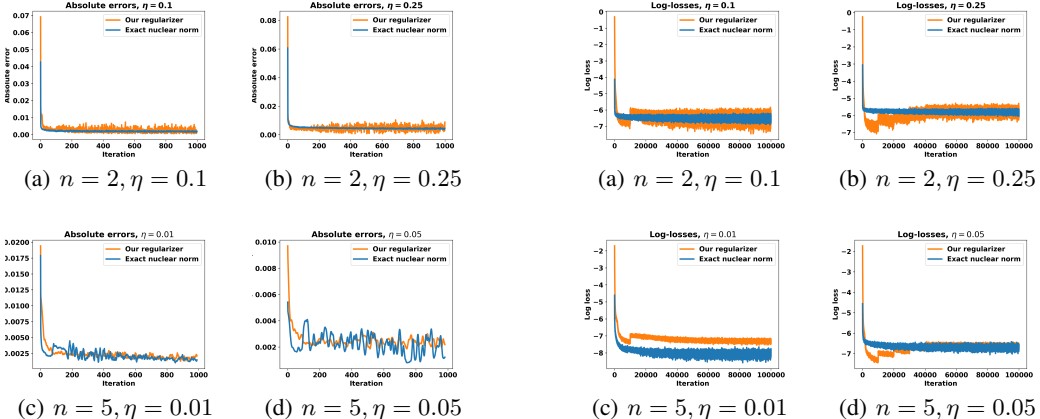

(a) $n = 2, \eta = 0.1$    (b) $n = 2, \eta = 0.25$      (a) $n = 2, \eta = 0.1$    (b) $n = 2, \eta = 0.25$

(c) $n = 5, \eta = 0.01$    (d) $n = 5, \eta = 0.05$      (c) $n = 5, \eta = 0.01$    (d) $n = 5, \eta = 0.05$

Figure 2: Mean absolute error of neural solutions to (8) (blue) and (9) (orange). Our regularizer obtains solutions with accuracy comparable to directly penalizing the Jacobian nuclear norm.

Figure 3: Log-objective values for (8) (blue) and (9) (orange) across training iterations. As predicted by Theorem 3.1, both problems converge to nearly identical objective values.

Figure 1 depicts the exact solution to (8) for $n = 2$ and two values of $\eta$, along with neural solutions to the same problem (8) and the problem with our regularizer (9). Solving our Jacobian-free problem (9) with 10 draws of $\epsilon$ per training iteration yields accurate solutions to the ROF problem for both values of $\eta$, with our problem yielding slightly diffuse transitions across the boundary of the unit disc.

Figure 2 confirms the accuracy our our method's solutions, which attain absolute error comparable to the neural solutions to (8).

Figure 3 depicts the objective values for Problems (8) and (9) on log scale across training iterations. As predicted by Theorem 3.1, both converge toward nearly identical objective values; the larger gap in Figure 3(c) is an artifact of the loss magnitudes being smaller and the plot being on log scale.

## 5  Applications

**Unsupervised denoising.**  In this section, we apply our regularizer $\mathcal{R}(x; f_\theta)$ to *unsupervised denoising*. We learn a denoiser $f_\theta$ that maps noisy images $x + \epsilon$ to clean images $x$ given a training set of noisy images *without* their corresponding clean images. We motivate the use of our regularizer via a connection with denoising by singular value shrinkage, and demonstrate that our unsupervised denoiser nearly matches the performance of a denoiser trained on clean-noisy image pairs.

**Singular value shrinkage.**  A line of work beginning with Shabalin and Nobel [2010] studies denoising by *singular value shrinkage* (SVS). These works seek to recover a low-rank data matrix $X \in \mathbb{R}^{D \times N}$ of *unknown* rank $r$ given only a single matrix $Y = X + \sigma_\epsilon Z$ of clean data $X$ corrupted by iid white noise $Z$. Since the clean data is low-rank, the components of $Y$ corresponding to its small singular values contain mostly noise, so SVS denoises $Y$ by applying a shrinkage function $\phi$ to its singular values $\sigma_d$. This function *shrinks* small singular values of $Y$ while leaving larger singular values untouched. For convenience, we denote the denoised matrix by $\phi(Y)$.

Gavish and Donoho [2017] show that under certain assumptions on the noise $Z$ and data $X$, one can derive the optimal shrinker $\phi$ that asymptotically minimizes $\|X - \phi(Y)\|_F^2$. In Appendix A.4, we show that this optimal shrinker is also the solution to the following problem:

$$\min_{A \in \mathbb{R}^{D \times D}} \frac{1}{2N} \|AY - Y\|_F^2 + \eta \|A\|_*, \tag{10}$$

where we set $\eta = \sigma_\epsilon^2$ to be equal to the noise variance. This problem is a special case of the following instance of Problem (3)

$$\inf_{f_\theta : \mathbb{R}^D \to \mathbb{R}^D} \mathbb{E}_{y \sim \mathcal{D}(\Omega)} \left[ \frac{1}{2} \|f_\theta(y) - y\|_2^2 + \eta \|Jf_\theta[y]\|_* \right] \tag{11}$$

when $\mathcal{D}(\Omega)$ is an empirical distribution over $N$ *noisy* training samples and $f_\theta$ is restricted to be a linear map. Just as (10) yields an optimal shrinker for denoising low-rank data which *globally* lies in a low-dimensional subspace, we conjecture that solving (11) yields an effective denoiser for manifold-supported data such as images, which *locally* lie near low-dimensional subspaces – even when trained on noisy images. We test this conjecture by solving (11) using a neural denoiser $f_\theta$. To make this problem tractable, we replace $\|Jf_\theta[y]\|_*$ with our regularizer $\mathcal{R}(y; f_\theta)$, which we compute using a single draw of $\epsilon$ per training iteration.

**Experiments.**  We train our denoiser by solving (11) with $\mathcal{D}(\Omega)$ being the empirical distribution over 288k *noisy* images from the Imagenet training set [Russakovsky et al., 2015]. Consequently,

**PSNR (dB) ↑**

| Method | Imagenet | | CBSD68 | |
| | $\sigma = 1$ | $\sigma = 2$ | $\sigma = 1$ | $\sigma = 2$ |
|---|---|---|---|---|
| BM3D | $21.26 \pm 2.81$ | $18.71 \pm 2.33$ | $19.42 \pm 1.88$ | $16.77 \pm 1.30$ |
| Ours | $23.10 \pm 3.12$ | $21.05 \pm 2.85$ | $21.08 \pm 2.04$ | $19.10 \pm 1.80$ |
| N2N | $23.12 \pm 3.05$ | $21.21 \pm 3.02$ | $20.37 \pm 1.71$ | $19.37 \pm 1.89$ |
| Supervised | $23.37 \pm 3.25$ | $21.39 \pm 2.97$ | $21.62 \pm 2.28$ | $19.54 \pm 1.95$ |

Table 1: Denoiser performance via average PSNR on held-out images. Our method performs nearly as well as a supervised denoiser, despite being trained exclusively on highly corrupted data without access to clean images.

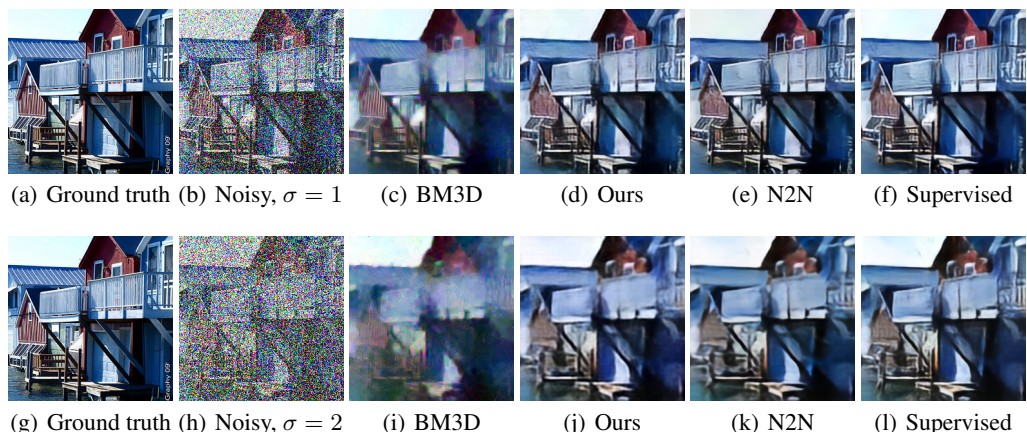

(a) Ground truth (b) Noisy, $\sigma = 1$ (c) BM3D (d) Ours (e) N2N (f) Supervised

(g) Ground truth (h) Noisy, $\sigma = 2$ (i) BM3D (j) Ours (k) N2N (l) Supervised

Figure 4: Denoiser performance comparison on held-out image corrupted by Gaussian noise with $\sigma = 1$ (first row) and $\sigma = 2$ (second row). Our method performs nearly as well as a supervised denoiser, despite being trained exclusively on highly corrupted data.

our denoiser does not see any clean images during training. The clean images' channel intensities lie in $[-1, 1]$, and we corrupt them with Gaussian noise with standard deviations $\sigma \in \{1, 2\}$. We set $\eta = \sigma^2$ when solving (11). We parametrize the denoiser $f_\theta = g_\theta \circ h_\theta$ as a Unet [Ronneberger et al., 2015], letting $h_\theta$ and $g_\theta$ be its downsampling and upsampling blocks, resp.

We benchmark our method against a supervised denoiser trained with the usual MSE loss $\|f_\theta(x + \epsilon) - x\|_2^2$ on clean-noisy pairs, Lehtinen et al. [2018]'s Noise2Noise (N2N) method, which requires access to independent noisy copies of each ground truth image during training, and BM3D, a classical unsupervised denoiser [Dabov et al., 2007]. We implement the supervised and N2N denoisers using the same Unet architecture as our denoiser, and train them on the same dataset with the same hyperparameters. We evaluate the denoisers via their average peak signal-to-noise ratio (PSNR) across the CBSD68 dataset [Martin et al., 2001] and across 100 randomly-drawn noisy images from the Imagenet validation set, randomly cropped to $256 \times 256$. We provide full details for this experiment in Appendix B.2.

We report each denoiser's performance in Table 1 and include 1-sigma error bars computed across the test images. Despite being trained *exclusively on highly corrupted images*, our denoiser nearly matches the performance of an ordinary supervised denoiser at both noise levels and performs comparably to Noise2Noise, which requires independent noisy copies of each ground truth image during training.

We further illustrate the comparison in Figure 4. All neural methods recover substantially more fine detail than the classical BM3D denoiser, particularly at the larger noise level $\sigma = 2$. Notably, our method performs nearly as well as a supervised denoiser, despite being trained exclusively on highly corrupted data.

We also demonstrate the sparsity-inducing effect of our regularizer on the singular values of $Jf_\theta$ in Figure 5, where we plot the Jacobian singular values of our denoiser and a supervised denoiser at a randomly-drawn validation image corrupted with $\sigma = 2$ Gaussian noise. We normalize the singular values so that each Jacobian's largest singular value is 1 and depict the singular values on log scale. As expected, our denoiser's Jacobian singular values decay more rapidly than those of the supervised denoiser at the same point.

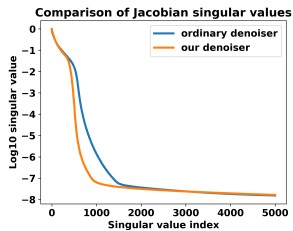

Figure 5: Jacobian singular values of supervised denoiser (blue) and our denoiser (orange) evaluated at a noisy held-out image with $\sigma = 2$.

These results show that our regularizer (6) can be used to construct a tractable non-linear generalization of Gavish and Donoho [2017]'s optimal shrinker that performs nearly as well as a supervised denoiser on image denoising tasks, despite being trained exclusively on highly corrupted images.

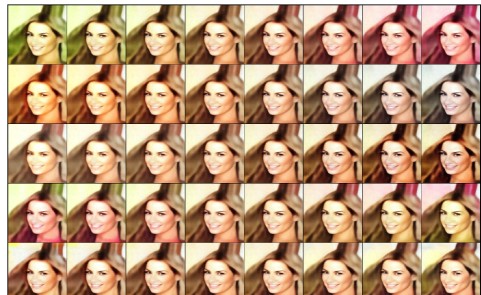

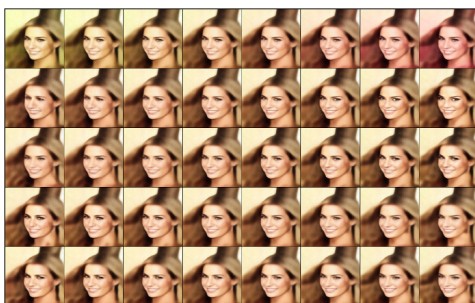

Figure 6: Traversals along Jacobian singular vectors of our *unregularized* encoder in latent space. These traversals edit the colors of the outputs but not other meaningful attributes.

Figure 7: Traversals along Jacobian singular vectors of our *regularized* encoder in latent space. Our regularizer enables these traversals to edit the subject's facial expressions.

**Representation learning.** We now apply our method to unsupervised representation learning. We train a deterministic autoencoder consisting of an encoder $f_\theta$ and a decoder $g_\phi$ on the CelebA dataset [Liu et al., 2015] and approximately penalize the Jacobian nuclear norm $\|J_{f_\theta}[x]\|_*$ of the encoder at training data $x \in \mathcal{D}(\Omega)$ using our regularizer $\mathcal{R}(x; f_\theta)$. This encourages the encoder to locally behave like a low-rank linear map whose *image* is low-dimensional. One may interpret this as a deterministic autoencoder with locally low-dimensional latent spaces. We demonstrate that the left-singular vectors of the encoder Jacobian $J_{f_\theta}[x]$ are semantically meaningful directions of variation about training data in latent space. We provide full experimental details in Appendix B.3.

To demonstrate our autoencoder's ability to learn meaningful representations, we select an arbitrary training point $x$ and traverse the latent space of our regularized autoencoder and an unregularized baseline along rays of the form $z = f_\theta(x) + \alpha u_\theta^d(x)$, where $u_\theta^d(x)$ is the $d$-th left-singular vector of the encoder Jacobian $J f_\theta[x]$. These left-singular vectors form a basis for the image of $J f_\theta[x]$ and approximate a basis for the tangent space of the latent manifold at $f_\theta(x)$. We depict decoded images along this traversal for our regularized autoencoder in Figure 7 and for the baseline in Figure 6.

The traversals in Figure 7 are semantically meaningful. For instance, a traversal along the first singular vector edits the tint of the decoded image, and a traversal along the second singular vector edits the facial expression of the image's subject. The traversals of the unregularized autoencoder's latent space in Figure 6 edit the colors of the decoded image but are unable to control other attributes.

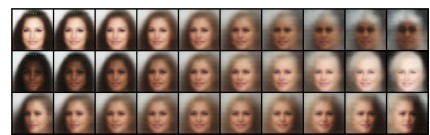

Figure 8: A $\beta$-VAE's latent traversals recover meaningful directions of variation, but the decoded images are highly diffuse.

We also follow Higgins et al. [2017] and visualize traversals along latent variables of a $\beta$-VAE at the same training point $x$ in Figure 8. While the $\beta$-VAE is able to discover meaningful directions of variation, the decoded images are highly diffuse, as is typical of VAEs. In contrast, our autoencoder's reconstructions retain finer details. We conjecture that our model's improved capacity results from our autoencoder's ability to learn a locally low-dimensional latent space without constraining its global structure.

## 6 Conclusion

The Jacobian nuclear norm $\|J f[x]\|_*$ is a natural regularizer for learning problems, where it steers solutions towards having low-rank Jacobians. Such functions are locally sensitive to changes in their inputs in only a few directions, which is an especially desirable prior for data that is supported on a low-dimensional manifold. However, computing $\|J f[x]\|_*$ naïvely requires both evaluating a Jacobian and taking its SVD; the combined cost of these operations is prohibitive for the high-dimensional maps $f$ that often arise in deep learning.

Our work resolves this computational challenge by generalizing a surprising result (2) from matrix learning to non-linear learning problems. As they rely on parametrizing the learned function $f = g \circ h$ as a composition of functions $g$ and $h$, our methods are tailor-made for deep learning, where such parametrizations are ubiquitous. We anticipate that the deep learning community will discover additional applications of Jacobian nuclear norm regularization to make use of our efficient methods.

As an efficient implementation of our regularizer relies on estimating the squared Jacobian Frobenius norm using Hutchinson's trace estimator, some error is inevitable. This error manifests itself in slightly diffuse boundaries in the solutions to the Rudin-Osher-Fatemi problem in Figure 1. However, we do not find this error problematic for high-dimensional applications such as unsupervised denoising as in Section 5. Future implementations of our method may employ more accurate estimators of the squared Jacobian Frobenius norm for applications where accuracy is of paramount concern.

## Acknowledgments and Disclosure of Funding

The MIT Geometric Data Processing Group acknowledges the generous support of Army Research Office grants W911NF2010168 and W911NF2110293, of National Science Foundation grant IIS-2335492, from the CSAIL Future of Data program, from the MIT–IBM Watson AI Laboratory, from the Wistron Corporation, and from the Toyota–CSAIL Joint Research Center.

Christopher Scarvelis acknowledges the support of the Natural Sciences and Engineering Research Council of Canada (NSERC), funding reference number CGSD3-557558-2021.

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

# A Proofs

## A.1 Proof of Equation (1)

We draw heavy inspiration from a proof by Fritz that appeared on MathOverflow. We will prove that

$$\|A\|_* = \min_{U,V:UV^\top=A} \frac{1}{2}\left(\|U\|_F^2 + \|V\|_F^2\right).$$

First suppose that $U, V$ are matrices such that $UV^\top = A$. By the matrix Hölder inequality,

$$\|A\|_* = \|UV^\top\|_* \le \|U\|_F \cdot \|V\|_F.$$

By the arithmetic mean-geometric mean (AM-GM) inequality,

$$\|U\|_F \cdot \|V\|_F = \sqrt{\|U\|_F^2 \cdot \|V\|_F^2} \le \frac{1}{2}\left(\|U\|_F^2 + \|V\|_F^2\right).$$

Combining these results, we obtain

$$\|A\|_* = \|UV^\top\|_* \le \inf_{U,V:UV^\top=A} \frac{1}{2}\left(\|U\|_F^2 + \|V\|_F^2\right).$$

To see that the inf is attained at $\|A\|_*$, take the compact SVD $A = U_A \Sigma V_A^\top$ and set $U := U_A\sqrt{\Sigma}, V := V_A\sqrt{\Sigma}$. Then clearly $UV^\top = A$, and

$$\frac{1}{2}\left(\|U\|_F^2 + \|V\|_F^2\right) = \frac{1}{2}\left(\|U_A\sqrt{\Sigma}\|_F^2 + \|V_A\sqrt{\Sigma}\|_F^2\right) = \frac{1}{2}\left(\|\sqrt{\Sigma}\|_F^2 + \|\sqrt{\Sigma}\|_F^2\right)$$

$$= \frac{1}{2}\left(\sum_{i=1}^r \sigma_i + \sum_{i=1}^r \sigma_i\right) = \|A\|_*$$

This proves Equation (1). An alternative proof using somewhat more complex methods is given in Mazumder et al. [2010]. ∎

## A.2 Proof of Theorem 3.1

Let $D(\Omega)$ be a data distribution with measure $\mu$ supported on a compact set $\Omega \subseteq \mathbb{R}^n$ that is absolutely continuously with respect to the Lebesgue measure $\lambda(\Omega)$, and let $\ell \in C^1(\mathbb{R}^m)$ be a continuously differentiable loss function. Let $C^\infty(\Omega)$ denote the set of infinitely differentiable functions on $\Omega$. We will show that:

$$\inf_{f\in C^\infty(\Omega)} \mathbb{E}_{x\sim\mathcal{D}(\Omega)}\left[\ell(f(x),x) + \eta\|Jf[x]\|_*\right]$$

$$= \inf_{\substack{h\in C^\infty(\Omega)\\g\in C^\infty(h(\Omega))}} \mathbb{E}_{x\sim\mathcal{D}(\Omega)}\left[\ell(g(h(x)),x) + \frac{\eta}{2}\left(\|Jg[h(x)]\|_F^2 + \|Jh[x]\|_F^2\right)\right]. \quad (12)$$

We denote the left-hand side objective by $E_L(f)$ and its inf by $(L)$; we denote the right-hand side objective by $E_R(g,h)$ and its inf by $(R)$. We prove that $(L) \le (R)$ and $(R) \le (L)$.

### A.2.1 $(L) \le (R)$

This is the easy direction. The basic observation is that if we parametrize $f = g \circ h$, then $Jf[x] = Jg[h(x)]Jh[x]$, and Srebro's identity (1) tells us that

$$\|Jf[x]\|_* = \min_{U,V:UV^\top=Jf[x]} \frac{1}{2}\left(\|U\|_F^2 + \|V\|_F^2\right) \le \frac{1}{2}\left(\|Jg[h(x)]\|_F^2 + \|Jh[x]\|_F^2\right)$$

The rest of the proof is book-keeping.

We first verify that $C^\infty(\Omega)$ is closed under composition by showing that:

$$C^\infty(\Omega) = \{g \circ h : h \in C^\infty(\Omega), g \in C^\infty(h(\Omega))\}. \tag{13}$$

The $\subseteq$ inclusion is straightforward: given any $f \in C^\infty(\Omega)$, just choose $h \equiv f$ and $g \equiv \mathrm{Id}$; these functions are clearly in the correct classes. The $\supseteq$ inclusion follows from the fact that the composition of $C^\infty$ functions is $C^\infty$ by the chain rule.

This yields:

$$\inf_{f \in C^\infty(\Omega)} E_L(f) = \inf_{\substack{h \in C^\infty(\Omega) \\ g \in C^\infty(h(\Omega))}} E_L(g \circ h).$$

We now use Srebro's identity as follows:

$$
\begin{aligned}
\inf_{\substack{h \in C^\infty(\Omega) \\ g \in C^\infty(h(\Omega))}} E_L(g \circ h) &= \inf_{\substack{h \in C^\infty(\Omega) \\ g \in C^\infty(h(\Omega))}} \mathbb{E}_{x \sim \mathcal{D}(\Omega)} \left[ \ell(g(h((x)), x) + \eta \| J(g \circ h)[x] \|_* \right] \\
&= \inf_{\substack{h \in C^\infty(\Omega) \\ g \in C^\infty(h(\Omega))}} \mathbb{E}_{x \sim \mathcal{D}(\Omega)} \left[ \ell(g(h((x)), x) + \eta \| Jg[h(x)] Jh[x] \|_* \right] \\
&\leq \inf_{\substack{h \in C^\infty(\Omega) \\ g \in C^\infty(h(\Omega))}} \mathbb{E}_{x \sim \mathcal{D}(\Omega)} \left[ \ell(g(h((x)), x) + \frac{\eta}{2} \left( \| Jg[h(x)] \|_F^2 + \| Jh[x] \|_F^2 \right) \right] \\
&= \inf_{\substack{h \in C^\infty(\Omega) \\ g \in C^\infty(h(\Omega))}} E_R(g \circ h).
\end{aligned}
$$

Consequently,

$$\inf_{f \in C^\infty(\Omega)} E_L(f) \leq \inf_{\substack{h \in C^\infty(\Omega) \\ g \in C^\infty(h(\Omega))}} E_R(g \circ h)$$

and hence $(L) \leq (R)$.

### A.2.2 $(R) \leq (L)$

This is the hard direction. The proof strategy is as follows:

1. We begin with a function $f_m \in C^\infty(\Omega)$ such that $E_L(f_m)$ is arbitrarily close to its inf over $C^\infty(\Omega)$. We use $f_m$ to construct parametric families of functions $g_m^z, h_m^z$ whose composition is a good local approximation to $f_m$ in a neighborhood of $z \in \Omega$, both pointwise and in terms of the energy contributions due to $x \in \Omega$ near $z$. This step relies crucially on our ability to construct optimal solutions to the RHS problem in (1).

2. We then stitch together these local approximations to form a sequence of global approximations $g_m^k, h_m^k$ to $f_m$. These functions are piecewise affine and hence not regular enough to lie in $C^\infty(\Omega)$ as required by the right-hand side of Equation (12).

3. Finally, mollifying the piecewise affine functions $g_m^k, h_m^k$ yields a minimizing sequence of $C^\infty(\Omega)$ functions $g_{m,\epsilon}^k, h_{m,\epsilon}^k$ such that $E_R(g_{m,\epsilon}^k, h_{m,\epsilon}^k)$ approaches the inf of $E_L$.

**Local approximations to $f_m$.** To begin, let $f_m \in C^\infty(\Omega)$ be a function that attains $E_L(f_m) \leq \inf_{f \in C^\infty(\Omega)} E_L(f) + \frac{1}{m}$. Fix some $z \in \Omega$, take the thin SVD of $Jf_m[z] = U_m(z) \Sigma_m(z) V_m(z)^\top$, and use it to define two parametric families of affine functions:

$$h_m^z(x) = \sqrt{\Sigma_m(z)} V_m(z)^\top x,$$

and

$$g_m^z(y) = U_m(z)\sqrt{\Sigma_m(z)}y + f_m(z) - Jf_m[z]z.$$

These functions satisfy two key properties:

$$g_m^z\left(h_m^z(x)\right) = f_m(z) + Jf_m[z](x - z) = f_m(x) + R_m^z(x), \tag{14}$$

where $\|R_m^z(x)\|_2 \in O(\|x - z\|_2^2)$ by Taylor's theorem, and

$$\frac{\eta}{2}\left(\|Jg_m^z[h_m^z(x)]\|_F^2 + \|Jh_m^z[x]\|_F^2\right) = \frac{\eta}{2}\left(\|\sqrt{\Sigma_m(z)}\|_F^2 + \|\sqrt{\Sigma_m(z)}\|_F^2\right) = \eta\|Jf_m[z]\|_*. \tag{15}$$

Using (14), we obtain

$$\ell(g_m^z(h_m^z(x)), x) = \ell(f_m(x) + R_m^z(x), x). \tag{16}$$

The continuity of $\ell$ ensures that we can make $\ell(f_m(x) + R_m^z(x), x)$ arbitrarily close to $\ell(f_m(x), x)$ by making $\|x - z\|_2$ sufficiently small.

Furthermore,

$$\|Jf_m[z]\|_* = \|Jf_m[x] + Jf_m[z] - Jf_m[x]\|_* \leq \|Jf_m[x]\|_* + \|Jf_m[z] - Jf_m[x]\|_*, \tag{17}$$

and as $f_m \in C^\infty$, $Jf_m$ is a continuous function, so we can make $\|Jf_m[z] - Jf_m[x]\|_*$ arbitrarily small by making $\|x - z\|$ sufficiently small.

The compositions of these functions $g_m^z, h_m^z$ are good local approximations to $f_m$ in a neighborhood of $z$, both pointwise (by (14)) and in terms of the energy contributions arising from $x \in \Omega$ near $z$ (by (16) and (17)).

**Global piecewise affine approximations to $f_m$.**  We now stitch together these local approximations to form a sequence of global approximations $g_m^k, h_m^k$ to $f_m$.

For each $k$, fix a set of points $Z_k = \{z_i\}_{i=1}^{N(k)}$ and use this set to partition $\Omega$ into Voronoi regions $V_i$. Choose the centroids $z_i$ such that $\max_{x \in V_i} \|x - z_i\|_2 \leq \epsilon_k$, for $\epsilon_k > 0$ sufficiently small to ensure that $|\|Jf_m[z_i]\|_* - \|Jf_m[x]\|_*| + |\ell(f_m(x) + R_m^{z_i}(x), x) - \ell(f_m(x), x)| < \frac{1}{k}$ for all $x \in V_i$ and for all regions $V_i$. (The compactness of $\Omega$ and the uniform continuity of $\|Jf_m\|_*$ on $\Omega$ and $\ell$ on its domain ensures that we can always find a finite set of centroids with this property.)

Then let $g_m^k, h_m^k$ be piecewise affine functions such that $g_m^k(x) := g_m^{z_i}(x)$ and $h_m^k(x) := h_m^{z_i}(x)$ for all $x \in \text{int}(V_i)$. For all points $x$ on a Voronoi boundary, define $g_m^k, h_m^k$ by averaging over the interiors of the Voronoi cells incident on the boundary. This yields:

$$\underset{x\sim\mathcal{D}(\Omega)}{\mathbb{E}}\left[\ell(g_m^k(h_m^k(x)),x)+\frac{\eta}{2}\left(\|Jg_m^k[h_m^k(x)]\|_F^2+\|Jh_m^k[x]\|_F^2\right)\right]$$

$$=\underset{x\sim\mathcal{D}(\Omega)}{\mathbb{E}}\left[\sum_{i=1}^{N(k)}[\ell(f_m(x)+R_m^{z_i}(x)),x)+\eta\|Jf_m[z_i]\|_*]\cdot\mathbb{1}\left[x\in V_i\right]\right]$$

$$\leq\underset{x\sim\mathcal{D}(\Omega)}{\mathbb{E}}\left[\sum_{i=1}^{N(k)}\left[\ell(f_m(x),x)+\eta\|Jf_m[x]\|_*+\frac{1}{k}\right]\cdot\mathbb{1}\left[x\in V_i\right]\right]$$

$$=\underset{x\sim\mathcal{D}(\Omega)}{\mathbb{E}}\left[\ell(f_m(x),x)+\eta\|Jf_m[x]\|_*+\frac{1}{k}\right]$$

$$=E_L(f_m)+\frac{1}{k}$$

$$\leq\inf_{f\in C^\infty(\Omega)}E_L(f)+\frac{1}{m}+\frac{1}{k}$$

and hence $E_R(g_m^k,h_m^k)\leq\inf_{f\in C^\infty(\Omega)}E_L(f)+\frac{1}{m}+\frac{1}{k}$.

**Mollifying the piecewise affine approximations.** The functions $g_m^k,h_m^k$ constructed in the previous section are piecewise affine and hence not regular enough to lie in $C^\infty(\Omega)$ as required by $(R)$. We now mollify these functions to yield a minimizing sequence of $C^\infty(\Omega)$ functions $g_{m,\epsilon}^k,h_{m,\epsilon}^k$ such that $E_R(g_{m,\epsilon}^k,h_{m,\epsilon}^k)$ approaches $\inf_{f\in C^\infty(\Omega)}E_L(f)$.

We mollify $g_m^k,h_m^k$ by convolving them against the standard mollifiers (infinitely differentiable and compactly supported on $B(0,\epsilon)$) to yield a sequence of $C^\infty(\Omega)$ functions $g_{m,\epsilon}^k,h_{m,\epsilon}^k$. We need to show that

$$E_R(g_{m,\epsilon}^k,h_{m,\epsilon}^k)\leq E_R(g_m^k,h_m^k)+\psi(\epsilon;m,k)$$

for some error $\psi(\epsilon;m,k)$ that vanishes as $\epsilon\to0$ for any $m,k$. We proceed by individually controlling the terms in $E_R(g_{m,\epsilon}^k,h_{m,\epsilon}^k)$.

**Controlling the error in** $\underset{x\sim\mathcal{D}(\Omega)}{\mathbb{E}}\left[\ell(g_{m,\epsilon}^k(h_{m,\epsilon}^k(x)),x)\right]$**.** We first show that it suffices to prove that $\|g_{m,\epsilon}^k\circ h_{m,\epsilon}^k-g_m^k\circ h_m^k\|_{L^1(\Omega,\mu)}\to0$ for any $m,k$. (Recall that $\mu$ is the measure associated with the data distribution $\mathcal{D}(\Omega)$.)

Note that as $\ell\in C^1(\mathbb{R}^m)$ and the image of the compact set $\Omega$ under the piecewise affine function $g_m^k\circ h_m^k$ is bounded, $\ell$ is $L$-Lipschitz on $g_m^k(h_m^k(\Omega))$. For any sequence of functions $f_n$ converging to $f$ in $L^1$, we then have:

$$\left|\int_\Omega\ell(f_n(x),x)d\mu-\int_\Omega\ell(f(x),x)d\mu\right|\leq\int_\Omega|\ell(f_n(x),x)-\ell(f(x),x)|\,d\mu$$

$$\leq\int_\Omega L\|f_n(x)-f(x)\|_2d\mu$$

$$=L\int_\Omega\|f_n(x)-f(x)\|_2d\mu$$

$$=L\|f_n-f\|_{L^1(\Omega,\mu)}$$

$$\underset{\epsilon\to0}{\to}0$$

It therefore suffices to show that $\|g_{m,\epsilon}^k\circ h_{m,\epsilon}^k-g_m^k\circ h_m^k\|_{L^1(\Omega,\mu)}\to0$ for any $m,k$. To this end, we will use the bound

$$\|g_{m,\epsilon}^k \circ h_{m,\epsilon}^k - g_m^k \circ h_m^k\|_{L^1(\Omega,\mu)} \le \|g_{m,\epsilon}^k \circ h_{m,\epsilon}^k - g_{m,\epsilon}^k \circ h_m^k\|_{L^1(\Omega,\mu)} + \|g_{m,\epsilon}^k \circ h_m^k - g_m^k \circ h_m^k\|_{L^1(\Omega,\mu)}$$

and show that each of the RHS terms goes to 0 as $\epsilon \to 0$.

We begin by controlling $\|g_{m,\epsilon}^k \circ h_{m,\epsilon}^k - g_{m,\epsilon}^k \circ h_m^k\|_{L^1(\Omega,\mu)}$. The obvious approach is to use the fact that $g_{m,\epsilon}^k$ is Lipschitz on the bounded domain $h_m^k(\Omega)$ and that $\|h_{m,\epsilon}^k - h_m^k\|_{L^1(\Omega,\mu)} \to 0$ by standard properties of mollifiers. However, this naïve approach fails because the Lipschitz constant of $g_{m,\epsilon}^k$ increases as $\epsilon \to 0$: We need to bound it in terms of a quantity independent of $\epsilon$. We will instead use the fact that the un-mollified function $g_m^k$ is piecewise affine and hence locally Lipschitz on the interior of each affine region.

Before proceeding with the remainder of the proof, we make a key observation. Given a Voronoi partition of the compact domain $\Omega$ into $N(k)$ cells $V_i$, we constructed $g_m^k, h_m^k$ as piecewise affine functions such that $g_m^k(x) := g_m^{z_i}(x)$ and $h_m^k(x) := h_m^{z_i}(x)$ for all $x \in \text{int}(V_i)$ – and for all points $x$ on a Voronoi boundary, we defined $g_m^k, h_m^k$ by averaging over the interiors of the Voronoi cells incident on the boundary. The affine functions $g_m^{z_i}, h_m^{z_i}$ were constructed to have the following properties:

$$g_m^z(h_m^z(x)) = f_m(z) + Jf_m[z](x - z) = f_m(x) + R_m^z(x), \tag{18}$$

where $\|R_m^z(x)\|_2 \in O(\|x - z\|_2^2)$ by Taylor's theorem, and

$$\frac{\eta}{2}\left(\|Jg_m^z[h_m^z(x)]\|_F^2 + \|Jh_m^z[x]\|_F^2\right) = \frac{\eta}{2}\left(\|\sqrt{\Sigma_m(z)}\|_F^2 + \|\sqrt{\Sigma_m(z)}\|_F^2\right) = \eta\|Jf_m[z]\|_*. \tag{19}$$

At any $x$ on the interior of a Voronoi cell, $g_m^k$ and $h_m^k$ are equal to some affine functions $g_m^{z_i}, h_m^{z_i}$, so by (19), $\frac{\eta}{2}\left(\|Jg_m^k[h_m^k(x)]\|_F^2 + \|Jh_m^k[x]\|_F^2\right) = \eta\|Jf_m[z_i]\|_* < +\infty$. In particular, $\|Jg_m^k[h_m^k(x)]\|_F^2$ and $\|Jh_m^k[x]\|_F^2$ must both be well-defined and finite at this $x$. This can only happen if $h_m^k(x)$ lies on the interior of a Voronoi cell, since otherwise $Jg_m^k[h_m^k(x)]$ would be undefined. Hence $h_m^k$ maps the interior of Voronoi cells to the interior of Voronoi cells; the contrapositive is that if $h_m^k(x)$ lies on a Voronoi boundary, then $x$ also lies on a Voronoi boundary. It follows that the preimage of the Voronoi boundaries under $h_m^k$ is a set of Lebesgue measure zero. Under the absolute continuity hypothesis, this is also a set of $\mu$-measure zero. We use this fact extensively throughout the remainder of the proof.

For any $m, k$, let $S_m^k \subseteq \Omega$ be the set of Voronoi boundaries from the partition we constructed earlier; this Voronoi partition depends on $m, k$ but not on $\epsilon$. We then begin by stripping the Voronoi boundaries from $\Omega$ to obtain $\Omega(m,k) := \Omega \setminus S_m^k$. This only removes a set of $\mu$-measure 0 from $\Omega$ and hence doesn't impact the remaining convergence results. Given $\epsilon$, we then partition $\Omega(m,k)$ into two subsets:

- Let $\Omega_0(m,k,\epsilon) \subseteq \Omega(m,k)$ be the set of $x \in \Omega(m,k)$ such that $d(x, S_m^k) > \epsilon$. (We use $d(p, S)$ to denote the distance from the point $p$ from the set $S$.)

- Let $\Omega_1(m,k,\epsilon)$ be its complement in $\Omega(m,k)$: the set of $x \in \Omega(m,k)$ such that $d(x, S_m^k) \le \epsilon$.

$\Omega_0(m,k,\epsilon)$ will be our good set, and $\Omega_1(m,k,\epsilon)$ will be a bad set whose measure converges to 0 as $\epsilon \to 0$. We decompose $\|g_{m,\epsilon}^k \circ h_{m,\epsilon}^k - g_{m,\epsilon}^k \circ h_m^k\|_{L^1(\Omega,\mu)}$ as follows:

$$\|g_{m,\epsilon}^k \circ h_{m,\epsilon}^k - g_{m,\epsilon}^k \circ h_m^k\|_{L^1(\Omega,\mu)} = \int_{\Omega_0(m,k,\epsilon)} \|g_{m,\epsilon}^k(h_{m,\epsilon}^k(x)) - g_{m,\epsilon}^k(h_m^k(x))\|_2 d\mu$$

$$+ \int_{\Omega_1(m,k,\epsilon)} \|g_{m,\epsilon}^k(h_{m,\epsilon}^k(x)) - g_{m,\epsilon}^k(h_m^k(x))\|_2 d\mu$$

We begin by showing that $\|g_{m,\epsilon}^k(h_{m,\epsilon}^k(x)) - g_{m,\epsilon}^k(h_m^k(x))\|_2 = 0$ for all $x \in \Omega_0(m, k, \epsilon)$. To this end, first recall that the standard mollifier supported on $B(0, \epsilon)$ is defined as follows:

$$\eta_\epsilon(y) = \begin{cases} C(\epsilon) \exp\left(\frac{1}{\|\frac{y}{\epsilon}\|_2^2 - 1}\right) & \text{for } \|y\|_2 \leq 1 \\ 0 & \text{for } \|y\|_2 > 1 \end{cases} \tag{20}$$

where $C(\epsilon) > 0$ is chosen to ensure that $\eta_\epsilon$ integrates to 1. Now, if $x \in \Omega_0(m, k, \epsilon)$, then $d(x, S_m^k) > \epsilon$ and hence $B(x, \epsilon)$ is entirely contained in the Voronoi cell $V_i$ containing $x$. (Note that $x$ cannot lie on a Voronoi boundary, as we have stripped the $\mu$-measure zero set of Voronoi boundaries $S_m^k$ from $\Omega$ before constructing $\Omega_0(m, k, \epsilon)$ and $\Omega_1(m, k, \epsilon)$.) On this ball $B(x, \epsilon)$, the linearity of $h_m^k(y) := \sqrt{\Sigma_m(z_i)} V_m(z_i)^\top y$ and the rotational symmetry of the mollifier $\eta_\epsilon$ yields:

$$h_{m,\epsilon}^k(x) = \int_{B(0,\epsilon)} h_m^k(y) \eta_\epsilon(x - y) dy \tag{21}$$

$$= \int_{B(0,\epsilon)} \sqrt{\Sigma_m(z_i)} V_m(z_i)^\top y \eta_\epsilon(x - y) dy \tag{22}$$

$$= \sqrt{\Sigma_m(z_i)} V_m(z_i)^\top \underbrace{\int_{B(0,\epsilon)} y \eta_\epsilon(x - y) dy}_{=x} \tag{23}$$

$$= \sqrt{\Sigma_m(z_i)} V_m(z_i)^\top y \tag{24}$$

$$= h_m^k(x). \tag{25}$$

Hence for all $x \in \Omega_0(m, k, \epsilon)$, $h_{m,\epsilon}^k(x) = h_m^k(x)$, and it follows that $g_{m,\epsilon}^k(h_{m,\epsilon}^k(x)) = g_{m,\epsilon}^k(h_m^k(x))$ and therefore $\|g_{m,\epsilon}^k(h_{m,\epsilon}^k(x)) - g_{m,\epsilon}^k(h_m^k(x))\|_2 = 0$. We conclude that

$$\int_{\Omega_0(m,k,\epsilon)} \|g_{m,\epsilon}^k(h_{m,\epsilon}^k(x)) - g_{m,\epsilon}^k(h_m^k(x))\|_2 d\mu = 0.$$

To bound the second term, note that the following inequalities hold $\mu$-almost everywhere on $\Omega$:

$$\|g_{m,\epsilon}^k(h_{m,\epsilon}^k(x)) - g_{m,\epsilon}^k(h_m^k(x))\|_2 \leq 2 \sup_{y \in h_m^k(\Omega(m,k))} \|g_{m,\epsilon}^k(y)\|_2 \leq 2 \sup_{y \in h_m^k(\Omega(m,k))} \|g_m^k(y)\|_2 =: M(m, k). \tag{26}$$

The second inequality can be derived using Jensen's inequality by first showing the following for all $y \in h_m^k(\Omega(m, k))$:

$$\|g_{m,\epsilon}^k(y)\|_2 = \|\int_{B(y,\epsilon)} g_m^k(z) \underbrace{\eta_\epsilon(z) dz}_{:=d\eta_\epsilon(z)}\|_2$$

$$\underset{\text{Jensen}}{\leq} \int_{B(y,\epsilon)} \|g_m^k(z)\|_2 d\eta_\epsilon(z)$$

$$\leq \sup_{y \in h_m^k(\Omega(m,k))} \|g_m^k(y)\|_2 \underbrace{\int_{B(y,\epsilon)} d\eta_\epsilon(z)}_{=1}$$

$$= \sup_{y \in h_m^k(\Omega(m,k))} \|g_m^k(y)\|_2,$$

which implies that

$$\sup_{y \in h_m^k(\Omega(m,k))} \|g_{m,\epsilon}^k(y)\|_2 \leq \sup_{y \in h_m^k(\Omega(m,k))} \|g_m^k(y)\|_2.$$

Returning to (26), we can bound the integral over the bad set $\int_{\Omega_1(m,k,\epsilon)} \|g_{m,\epsilon}^k(h_{m,\epsilon}^k(x)) - g_{m,\epsilon}^k(h_m^k(x))\|_2 d\mu$ as follows:

$$\int_{\Omega_1(m,k,\epsilon)} \|g_{m,\epsilon}^k(h_{m,\epsilon}^k(x)) - g_{m,\epsilon}^k(h_m^k(x))\|_2 d\mu \leq M(m,k) \cdot \mu(\Omega_1(m,k,\epsilon))$$

where $\mu(\Omega_1(m,k,\epsilon))$ is the measure of $\Omega_1(m,k,\epsilon)$. We will now show that $\mu(\Omega_1(m,k,\epsilon)) \to 0$, which will imply

$$\int_{\Omega_1(m,k,\epsilon)} \|g_{m,\epsilon}^k(h_{m,\epsilon}^k(x)) - g_{m,\epsilon}^k(h_m^k(x))\|_2 d\mu \xrightarrow[\epsilon \to 0]{} 0,$$

and therefore allow us to conclude that

$$\|g_{m,\epsilon}^k \circ h_{m,\epsilon}^k - g_{m,\epsilon}^k \circ h_m^k\|_{L^1(\Omega,\mu)}^2 \xrightarrow[\epsilon \to 0]{} 0.$$

**Proving that $\mu(\Omega_1(m,k,\epsilon)) \to 0$.**   Recall that:

$$\Omega_1(m,k,\epsilon) := \left\{ x \in \Omega(m,k) : d(x, S_m^k) \leq \epsilon \right\}, \tag{27}$$

where $d(x, S_m^k)$ denotes the distance from the point $x$ to the union of Voronoi boundaries $S_m^k$. Note that this set is a union of cylinders of radius $\epsilon$ centered at the Voronoi boundaries $S_m^k$. As $S_m^k$ has Lebesgue measure 0, the Lebesgue measure of the cylinders $B(m,k,\epsilon)$ also goes to 0 as the radius $\epsilon \to 0$. The absolute continuity of $\mu$ then implies that $\mu(\Omega_1(m,k,\epsilon)) \to 0$ as well.

Using these results, we obtain

$$\int_{\Omega_1(m,k,\epsilon)} \|g_{m,\epsilon}^k(h_{m,\epsilon}^k(x)) - g_{m,\epsilon}^k(h_m^k(x))\|_2 d\mu \xrightarrow[\epsilon \to 0]{} 0,$$

and therefore conclude that

$$\|g_{m,\epsilon}^k \circ h_{m,\epsilon}^k - g_{m,\epsilon}^k \circ h_m^k\|_{L^1(\Omega,\mu)}^2 \xrightarrow[\epsilon \to 0]{} 0.$$

This completes the proof that $\|g_{m,\epsilon}^k \circ h_{m,\epsilon}^k - g_{m,\epsilon}^k \circ h_m^k\|_{L^1(\Omega,\mu)} \to 0$ as $\epsilon \to 0$.

Controlling the second term $\|g_{m,\epsilon}^k \circ h_m^k - g_m^k \circ h_m^k\|_{L^1(\Omega,\mu)}$ is easier. Indeed, $g_{m,\epsilon}^k \circ h_m^k \to g_m^k \circ h_m^k$ pointwise a.e. as $\epsilon \to 0$ by standard properties of mollifiers. Furthermore, the sequence $g_{m,\epsilon}^k \circ h_m^k$ is dominated almost everywhere by the function identically equal (componentwise) to $\max_{x \in \Omega(m,k)} g_m^k(h_m^k(x))$; this function is in $L^1(\Omega,\mu)$ because $\Omega$ is a compact domain. It follows from the dominated convergence theorem that $\|g_{m,\epsilon}^k \circ h_m^k - g_m^k \circ h_m^k\|_{L^1(\Omega,\mu)} \to 0$ as $\epsilon \to 0$.

We have shown that each of the following RHS terms goes to 0 as $\epsilon \to 0$:

$$\|g_{m,\epsilon}^k \circ h_{m,\epsilon}^k - g_m^k \circ h_m^k\|_{L^1(\Omega,\mu)} \leq \|g_{m,\epsilon}^k \circ h_{m,\epsilon}^k - g_{m,\epsilon}^k \circ h_m^k\|_{L^1(\Omega,\mu)} + \|g_{m,\epsilon}^k \circ h_m^k - g_m^k \circ h_m^k\|_{L^1(\Omega,\mu)},$$

which allows us to conclude that $\|g_{m,\epsilon}^k \circ h_{m,\epsilon}^k - g_m^k \circ h_m^k\|_{L^1(\Omega,\mu)} \to 0$ and consequently

$$\left| \mathbb{E}_{x \sim \mathcal{D}(\Omega)} \left[ \ell(g_{m,\epsilon}^k(h_{m,\epsilon}^k(x)), x) \right] - \mathbb{E}_{x \sim \mathcal{D}(\Omega)} \left[ \ell(g_m^k(h_m^k(x)), x) \right] \right| \xrightarrow[\epsilon \to 0]{} 0$$

as $\epsilon \to 0$, for any $m, k$. It remains to prove similar convergence results for the remaining terms in $E_R$.

**Controlling the error in** $\frac{\eta}{2} \underset{x \sim \mathcal{D}(\Omega)}{\mathbb{E}} \left[ \|Jg_{m,\epsilon}^k[h_{m,\epsilon}^k(x)]\|_F^2 + \|Jh_{m,\epsilon}^k[x]\|_F^2 \right]$. We now show that $\int_\Omega \|Jg_{m,\epsilon}^k[h_{m,\epsilon}^k(x)]\|_F^2 d\mu \to \int_\Omega \|Jg_m^k[h_m^k(x)]\|_F^2 d\mu$ via the dominated convergence theorem (DCT). First note that $\Omega(m, k)$ and $\Omega$ only differ by a set of $\mu$-measure zero (the Voronoi boundaries $S_m^k$), so we can equivalently prove

$$\int_{\Omega(m,k)} \|Jg_{m,\epsilon}^k[h_{m,\epsilon}^k(x)]\|_F^2 d\mu \underset{\epsilon \to 0}{\to} \int_{\Omega(m,k)} \|Jg_m^k[h_m^k(x)]\|_F^2 d\mu.$$

This allows us to avoid points $x$ such that $x$ or $h_m^k(x)$ lie on Voronoi boundaries; these are problematic because if $h_m^k(x)$ lies on a Voronoi boundary, then $Jg_m^k[h_m^k(x)]$ is undefined.

First note that as $g_m^k, h_m^k$ are piecewise affine and the mollifiers are compactly supported, for any given $x \in \Omega(m, k)$, one can choose $\epsilon > 0$ sufficiently small so that $g_{m,\epsilon}^k[h_{m,\epsilon}^k(x)] = g_m^k[h_m^k(x)]$ and hence $\|Jg_{m,\epsilon}^k[h_{m,\epsilon}^k(x)]\|_F^2 = \|Jg_m^k[h_m^k(x)]\|_F^2$.

In particular, for any $x \in \Omega(m, k)$, let $\epsilon_1 < d(x, S_m^k)$ and $\epsilon_2 < d(h_m^k(x), S_m^k)$; these can both be $> 0$ because $d(x, S_m^k) > 0, d(h_m^k(x), S_m^k) > 0$ for $x \in \Omega(m, k)$. Then define $\epsilon := \min\{\epsilon_1, \epsilon_2\}$. (21) shows why choosing $\epsilon_1 < d(x, S_m^k)$ implies $h_{m,\epsilon}^k(x) = h_m^k(x)$; similar arguments hold show that $\epsilon_2 < d(h_m^k(x), S_m^k)$ implies $g_{m,\epsilon}^k(h_m^k(x)) = g_m^k(h_m^k(x))$. We then have $g_{m,\epsilon}^k[h_{m,\epsilon}^k(x)] = g_{m,\epsilon}^k[h_m^k(x)] = g_m^k[h_{m,\epsilon}^k(x)]$ as desired.

Hence for this choice of $\epsilon$, $Jg_{m,\epsilon}^k[h_{m,\epsilon}^k(x)] = Jg_m^k[h_m^k(x)]$ and consequently $\|Jg_{m,\epsilon}^k[h_{m,\epsilon}^k(x)]\|_F^2 = \|Jg_m^k[h_m^k(x)]\|_F^2$. (Recall that $Jg_m^k[h_m^k(x)]$ is well-defined for $x \in \Omega(m, k)$.) It follows that $\|Jg_{m,\epsilon}^k[h_{m,\epsilon}^k(x)]\|_F^2 \to \|Jg_m^k[h_m^k(x)]\|_F^2$ pointwise on $\Omega(m, k)$ and pointwise $\mu$-ae on $\Omega$.

Furthermore, another argument via Jensen's inequality shows that $\|Jg_{m,\epsilon}^k[h_{m,\epsilon}^k(x)]\|_F^2$ is dominated by the function $A_m^k$ that is identically equal to $\sup_{x \in \Omega(m,k)} \|Jg_m^k[h_m^k(x)]\|_F^2$; this function is integrable because $\Omega$ is compact.

The DCT then lets us conclude that $\|Jg_{m,\epsilon}^k[h_{m,\epsilon}^k(x)]\|_F^2 \to \|Jg_m^k[h_m^k(x)]\|_F^2$ in $L^1$ and hence that

$$\int_\Omega \|Jg_{m,\epsilon}^k[h_{m,\epsilon}^k(x)]\|_F^2 d\mu \underset{\epsilon \to 0}{\to} \int_\Omega \|Jg_m^k[h_m^k(x)]\|_F^2 d\mu.$$

The same argument also allows us to conclude that

$$\int_\Omega \|Jh_{m,\epsilon}^k(x)]\|_F^2 d\mu \underset{\epsilon \to 0}{\to} \int_\Omega \|Jh_m^k(x)]\|_F^2 d\mu.$$

We have by now shown that

$$E_R(g_{m,\epsilon}^k, h_{m,\epsilon}^k) \le E_R(g_m^k, h_m^k) + \psi(\epsilon; m, k)$$

for some $\psi(\epsilon; m, k) \to 0$ as $\epsilon \to 0$. Combining this with our earlier results, we get

$$E_R(g_{m,\epsilon}^k, h_{m,\epsilon}^k) \le E_R(g_m^k, h_m^k) + \psi(\epsilon; m, k) \le \inf_{f \in C^\infty(\Omega)} E_L(f) + \frac{1}{m} + \frac{1}{k} + \psi(\epsilon; m, k).$$

As we can make $\frac{1}{m} + \frac{1}{k} + \psi(\epsilon; m, k)$ arbitrarily small by first choosing $m, k$ sufficiently large and then choosing $\epsilon(m, k) > 0$ to make $\psi(\epsilon; m, k)$ sufficiently small, we can finally conclude that $(R) \le (L)$ as desired.

This completes the proof of the theorem. ∎

## A.3 Proof of Theorem 3.2

We will show that

$$\sigma^2 \|Jf[x]\|_F^2 = \mathop{\mathbb{E}}_{\epsilon \sim \mathcal{N}(0,\sigma^2 I)} \left[ \|f(x+\epsilon) - f(x)\|_2^2 \right] + O(\sigma^2).$$

Since $f : \mathbb{R}^n \to \mathbb{R}^m$ is continuously differentiable, Taylor's theorem states that:

$$f(x+\epsilon) = f(x) + Jf[x]\epsilon + R(x+\epsilon),$$

where $\|R(x+\epsilon)\|_2 \in O(\|\epsilon\|_2^2)$. Rearranging, taking squared Euclidean norms, and expanding the square, we obtain:

$$
\begin{aligned}
\|Jf[x]\epsilon\|_2^2 &= \|f(x+\epsilon) - f(x) - R(x+\epsilon)\|_2^2 \\
&= \|f(x+\epsilon) - f(x)\|_2^2 - 2 \cdot \langle f(x+\epsilon) - f(x), R(x+\epsilon) \rangle + \underbrace{\|R(x+\epsilon)\|_2^2}_{\in O(\|\epsilon\|_2^4)} \\
&\leq \|f(x+\epsilon) - f(x)\|_2^2 + 2 \underbrace{\|R(x+\epsilon)\|_2}_{\in O(\|\epsilon\|_2^2)} \cdot \underbrace{\|f(x+\epsilon) - f(x)\|_2}_{\in O(\|\epsilon\|_2)} + O(\|\epsilon\|_2^4) \\
&= \|f(x+\epsilon) - f(x)\|_2^2 + O(\|\epsilon\|_2^3) \\
&= \|f(x+\epsilon) - f(x)\|_2^2 + O(\|\epsilon\|_2^2).
\end{aligned}
$$

Hutchinson's trace estimator implies that for any matrix $A$, $\|A\|_F^2 = \mathbb{E}_{\epsilon \sim \mathcal{N}(0,I)}[\|A\epsilon\|_2^2]$. In particular,

$$
\begin{aligned}
\sigma^2 \|Jf[x]\|_F^2 &= \mathop{\mathbb{E}}_{\epsilon \sim \mathcal{N}(0,\sigma^2 I)} \left[ \|Jf[x]\epsilon\|_2^2 \right] \\
&= \mathop{\mathbb{E}}_{\epsilon \sim \mathcal{N}(0,\sigma^2 I)} \left[ \|f(x+\epsilon) - f(x)\|_2^2 + O(\|\epsilon\|_2^3) \right] \\
&= \mathop{\mathbb{E}}_{\epsilon \sim \mathcal{N}(0,\sigma^2 I)} \left[ \|f(x+\epsilon) - f(x)\|_2^2 + O(\underbrace{\mathbb{E}\|\epsilon\|_2^2}_{=\sigma^2 n}) \right. \\
&= \mathop{\mathbb{E}}_{\epsilon \sim \mathcal{N}(0,\sigma^2 I)} \left[ \|f(x+\epsilon) - f(x)\|_2^2 \right] + O(\sigma^2),
\end{aligned}
$$

which completes the proof of the result. ∎

## A.4 Optimal shrinkage via nuclear norm regularization

Let $X \in R^{D \times N}$ be a low-rank matrix of clean data and $Y = X + \sigma_\epsilon Z$ be a matrix of data corrupted by iid white noise $Z$. In this appendix, we show that the solution to

$$\min_{A \in \mathbb{R}^{D \times D}} \frac{1}{2N} \|AY - Y\|_F^2 + \eta \|A\|_*, \tag{28}$$

coincides with Gavish and Donoho [2017]'s optimal shrinker for the squared Frobenius norm loss when $\eta$ is set to the noise variance $\sigma_\epsilon^2$ and as the "aspect ratio" $\beta := \frac{d}{n} \to 0$.

$A^*$ is optimal for problem (28) iff $0 \in \partial \phi(A^*)$. Using well-known results from convex optimization, this condition is equivalent to:

$$\frac{1}{N\eta}(Y - A^*Y)Y^\top \in \partial \| \cdot \|_*(A^*). \tag{29}$$

Furthermore, the subgradient of the nuclear norm is:

$$\partial \| \cdot \|_*(A) = \left\{ U_A V_A^\top + W : U_A^\top W = 0, W V_A = 0, \sigma_{\max}(W) \leq 1 \right\}, \tag{30}$$

where $A = U_A \Sigma_A V_A^\top$ is an SVD of $A$. We will show that the solution to (28) is

$$A^* = U \Gamma U^\top, \tag{31}$$

where $Y = U \Sigma V^\top$ is an SVD of the noisy data matrix, and

$$\Gamma_d = \begin{cases} 1 - \frac{N\eta}{\sigma_d^2}, & \sigma_d \geq \sqrt{N\eta} \\ 0, & \sigma_d \leq \sqrt{N\eta}. \end{cases} \tag{32}$$

The idea is to use the SVD $Y = U \Sigma V^\top$ and the ansatz $A^* = U \Gamma U^\top$ to rewrite the LHS of the inclusion (29) as follows:

$$\frac{1}{N\eta}(Y - A^* Y) Y^\top = U \left( \frac{1}{N\eta} (I - \Gamma) \Sigma^2 \right) U^\top.$$

We then express the middle diagonal term as follows:

$$\frac{1}{N\eta}(I - \Gamma)\Sigma^2 = I_T + \frac{1}{N\eta}(I - \Gamma)\Sigma^2 - I_T$$

where $I_T$ is the identity matrix with all columns $\geq T$ set to zero (i.e. an orthogonal projection matrix onto the first $T$ coordinates). This then yields

$$U \left( \frac{1}{N\eta}(I - \Gamma)\Sigma^2 \right) U^\top = U_T U_T^\top + U \left( \frac{1}{N\eta}(I - \Gamma)\Sigma^2 - I_T \right) U^\top,$$

where $U_T = U I_T$ is $U$ with all columns $\geq T$ set to 0, and $T$ is the first index such that $\sigma_T \leq \sqrt{N\eta}$. As the optimal $\Gamma$ in (32) sets all entries corresponding to singular values $\sigma_d \leq \sqrt{N\eta}$ to zero, we can in fact rewrite $A^* = (U I_T)\Gamma(U I_T)^\top$. It follows that $U_T U_T^\top = U I_T (U I_T)^\top$ is also of the form $U_A V_A^\top$ for a valid SVD of $A^*$ (the $U I_T$ can serve as both left- and right-singular vectors).

We have therefore expressed the LHS of the inclusion (29) as:

$$\frac{1}{N\eta}(Y - A^* Y) Y^\top = U_A V_A^\top + W$$

for $U_A V_A^\top = U_T U_T^\top$ and $W = U \left( \frac{1}{N\eta}(I - \Gamma)\Sigma^2 - I_T \right) U^\top$. This $W$ satisfies all of the conditions in (30).

Furthermore, if we apply this optimal $A^*$ to the data matrix $Y$, we obtain $A^* Y = U \Gamma \Sigma V^\top$, where

$$(\Gamma \Sigma)_d = \begin{cases} \frac{\sigma_d^2 - N\eta}{\sigma_d}, & \sigma_d^2 \geq N\eta \\ 0, & \sigma_d^2 \leq N\eta \end{cases} \tag{33}$$

If we set $\eta$ to be equal to the noise variance $\sigma_\epsilon^2$, then this agrees exactly with the optimal shrinker for the case $\beta := \frac{D}{N} \to 0$ from Gavish and Donoho [2017] under the same noise model $Y = X + \sigma_\epsilon Z$.

## B  Experimental details

### B.1  Validation experiments: Rudin-Osher-Fatemi (ROF) problem

**Architecture.**  In all ROF experiments, we parametrize $f_\theta = g_\theta \circ h_\theta$, where $g_\theta$ and $h_\theta$ are both two-layer MLPs with 100 hidden units. We apply a Fourier feature mapping [Tancik et al., 2020] to

the input coordinates $x$ before passing them through $h_\theta$. We use ELU activations in both neural nets and find the use of differentiable non-linearities to be crucial for obtaining accurate solutions.

**Training details.** We train all neural models using the AdamW optimizer [Loshchilov and Hutter, 2019] at a learning rate of $10^{-4}$ for 100,000 iterations with a batch size of 10,000. In the $n = 2$ case, we integrate over the box $[-10, 10]^2$, and in the $n = 5$ case, we integrate over the box $[-2, 2]^5$.

We employ a warmup strategy for solving our problem (9). We first train our neural nets at $\eta = 0.05$ in the $n = 2$ case and $\eta = 0.01$ in the $n = 5$ case for 10,000 iterations, and then increase $\eta$ by 0.05 and 0.01, respectively, each 10,000 iterations until we reach the desired value of $\eta$. We then continue training until we reach 100,000 total iterations.

Each training run for (8) takes approximately 2 hours, and each training run for (9) takes approximately 45 minutes on a single V100 GPU.

## B.2 Denoising

**Architecture.** Our architecture for all denoising models is based on the UNet implemented in the Github repository for Zhang et al. [2021]. Each model is of the form $f = g \circ h$, where $h$ consists of the head, downsampling blocks, and body block of the Unet, and $g$ consists of a repeated body block, the upsampling blocks, and the tail. We replace all ReLU activations with ELU but leave the remainder of the architecture unchanged.

**Training details.** All neural models are trained on 288,049 images from the ImageNet Large Scale Visual Recognition Challenge 2012 training set [Russakovsky et al., 2015] which we randomly crop and rescale to $128 \times 128$. The code for loading and pre-processing this training data is borrowed from Rombach et al. [2021].

We train all neural models using the AdamW optimizer [Loshchilov and Hutter, 2019] for 2 epochs at a learning rate of $10^{-4}$ then for a final epoch with learning rate $10^{-5}$. Each denoising model takes approximately 5 hours to train on a single V100 GPU.

The training objective for our denoiser is (11) with $\eta = \sigma$. The training objective for the supervised denoiser is the usual MSE loss:

$$\inf_{\substack{f_\theta : \mathbb{R}^D \to \mathbb{R}^D}} \mathbb{E}_{\substack{x \sim \mathcal{D}(\Omega) \\ \epsilon \sim \mathcal{N}(0, I)}} \left[ \frac{1}{2} \| f_\theta(x + \sigma\epsilon) - x \|_2^2 \right],$$

where $\mathcal{D}(\Omega)$ now denotes the empirical distribution over clean training images. The training objective for the Noise2Noise denoiser is:

$$\inf_{\substack{f_\theta : \mathbb{R}^D \to \mathbb{R}^D}} \mathbb{E}_{\substack{x \sim \mathcal{D}(\Omega) \\ \epsilon_1, \epsilon_2 \sim \mathcal{N}(0, I)}} \left[ \frac{1}{2} \| f_\theta(x + \sigma\epsilon_1) - (x + \sigma\epsilon_2) \|_2^2 \right].$$

Note that this requires access to independent noisy copies of the same clean image during training.

**Evaluation details.** We evaluate each denoiser by measuring their average peak signal-to-noise ratio (PSNR) in decibels (dB) on 100 randomly-drawn images from the ImageNet validation set, randomly cropped to $256 \times 256$. We corrupt each held-out image with Gaussian noise with the same standard deviation that the respective models were trained on ($\sigma \in \{1, 2\}$) and denoise them using each neural model along with BM3D [Dabov et al., 2007], a popular classical baseline for unsupervised denoising.

## B.3 Representation learning

We use the $\beta$-VAE implementation from the AntixK PyTorch-VAE repo and use the default hyperparameters (in particular, we set $\beta = 10$) but set the latent dimension to 32, as we find that this yields more meaningful latent traversals. Training this $\beta$-VAE takes approximately 30 minutes on a single V100 GPU.

We describe the architecture and training details for our regularized and unregularized autoencoder which we use to generate the latent traversals in Figures 6 and 7. Training this autoencoder with the de

**Deterministic autoencoder architecture.**  Our autoencoder operates on $256 \times 256$ images from the CelebA dataset. To reduce the memory and compute costs of our autoencoder, we perform a discrete cosine transform (DCT) using the `torch-dct` package and keep only the first 80 DCT coefficients. We then pass these coefficients into our autoencoder.

Our deterministic autoencoder consists of an encoder $f_\theta$ followed by a decoder $g_\phi$. The encoder $f_\theta$ is parametrized as a two-layer MLP with 10,000 hidden units; the latent space is 700-dimensional. The decoder $g_\phi$ consists of a two-layer MLP with 10,000 hidden units and $3 * 80 * 80 = 19200$ output dimensions, followed by an inverse DCT, and finally a UNet. We use the same UNet as in the denoising experiments described in Appendix B.2.

**Training details.**  We train our autoencoders with the following objective:

$$\inf_{f_\theta, g_\phi} \mathbb{E}_{x \sim \mathcal{D}(\Omega)} \left[ \frac{1}{2} \|g_\phi(f_\theta(x)) - x\|_2^2 + \eta \mathcal{R} x; (f_\theta) \right], \tag{34}$$

where $\mathcal{D}(\Omega)$ is the CelebA training set [Liu et al., 2015].

This is a standard deterministic autoencoder objective, with our regularizer approximating the Jacobian nuclear norm $\|Jf_\theta[x]\|_*$ of the encoder $f_\theta$.

We train the unregularized autoencoder with $\eta = 0$, and the regularized autoencoder with $\eta = 0.5$. In both cases, we train on the CelebA training set for 4 epochs using the AdamW optimizer [Loshchilov and Hutter, 2019] with a learning rate of $10^{-4}$. Training these autoencoders takes approximately 4 hours each on a single V100 GPU.

**Generating latent traversals.**  To generate the latent traversals for our autoencoder, we draw a point $x$ from the training set, compute the encoder Jacobian $Jf_\theta[x]$, and take its SVD to obtain $Jf_\theta[x] = U\Sigma V^\top$. We then take the first 5 left-singular vectors (i.e. the first 5 columns of $U$) and compute $z = f_\theta(x) + \alpha u_\theta^d(x)$, where $u_\theta^d(x)$ denotes the $d$-th column of $U$. Here $\alpha$ denotes a scalar coefficient; it ranges over $[-20000, 20000]$ for the unregularized autoencoder and $[-2000, 2000]$ for the regularized autoencoder.

To generate the latent traversals for the $\beta$-VAE, we encode the same training point $x$ and replace the $d$-th latent coordinate with an equispaced traversal of $[-3, 3]$ for $d \in \{1, 2, 11\}$; these are the first three meaningful latent traversals for this training point.

