# OpenReview forum: "Nuclear Norm Regularization for Deep Learning"
_NeurIPS.cc/2024/Conference — NeurIPS 2024 poster_

### Official Review · Reviewer_Gitn · 2024-07-02

**Soundness:** 4
**Presentation:** 4
**Contribution:** 4
**Rating:** 8
**Confidence:** 3

**Summary:**

The paper proposes a method to regularize the nuclear norm of the Jacobian of a function, e.g., one that represents a neural network. This method builds on prior art and includes the authors’ novel contribution as follows: The authors reference prior art for an equivalent problem formulation of nuclear norm regularization that avoids the computation of SVD. This equivalence also enables the authors’ novel result, which is that the nuclear norm of the Jacobian $Jf$ of a composite function $f=g \circ h $ is equal to the average of the squared Frobenius norms of $Jg$ and $Jh$; this result makes the proposed method perfectly apt for neural network training. The authors then use an approximation to the Frobenius norm of the Jacobians to avoid explicitly calculating a large Jacobian matrix, which significantly reduces computational cost and storage. The method is validated with a nuclear norm-regularized problem whose closed form solution is known. Finally, the efficacy of the method is shown on two applications: unsupervised denoising and representation learning.

**Strengths:**

-	The authors’ key finding on the Jacobian of a composite function is interesting,
original and significant.
-	The paper is clear and well-written.
-	The authors have given a good summary of the background and preliminaries.
-	The authors have clearly identified the parts of their method based on prior art and based on their own contribution.
-	The validation and application examples that demonstrate the proposed method’s efficacy are generally convincing.
-	The experiments of the paper seem reproducible.

**Weaknesses:**

Major comment:
- In the representation learning application, we only see the method’s efficacy on a single image. Could there be a way to quantify the performance of the method over a whole test dataset?

Minor comments:
- I found the following two references that also compute nuclear norm without the SVD. Could the authors either mention these references in their manuscript or clarify why these aren’t relevant?
[1] https://icml.cc/Conferences/2010/papers/196.pdf
[2] https://www.ijcai.org/proceedings/2017/0436.pdf

- Figure 1 has only the color legend. Could the authors confirm (and explicitly state on their manuscript) that the x- and y-axes correspond to each coordinate of the input x?
- In the caption of Figure 3, the authors say that “As predicted by Theorem 3.1, both problems converge to the same objective value.“ Could they use a less strong claim here? Possibly “nearly identical” as they said in the main text; otherwise, the gap in 3(c) is confusing.
- Could the authors reformat the subsection titles in Section 5 so that it is clear in a glance that “Unsupervised denoising” and “Representation learning” are the two application examples, and “Singular value shrinkage” and “Experiments” are under “Unsupervised denoising”?
- The PSNR values in Table 1 are based on averaging over only 100 images. Is there a reason why more images were not used? Could the authors repeat the experiment by averaging over more images?
- It is slightly confusing that the order of the methods in the Table 1 and Figure 4 do not match each other. Could the authors follow the same order?

**Questions:**

Please see Weaknesses.

**Limitations:**

The authors briefly comment on the limitation of their method regarding the approximation of the squared Frobenius norm of the Jacobian in Conclusions. I think this is sufficient.

---

> ### Author Rebuttal · Authors · 2024-08-06
>
> Thank you for your thoughtful review of our manuscript. We are glad you appreciate the originality and significance of our method for Jacobian nuclear norm regularization.
>
> *"In the representation learning application, we only see the method’s efficacy on a single image. Could there be a way to quantify the performance of the method over a whole test dataset?"*
>
> We depict latent traversals in our autoencoder's latent space for a single non-cherry-picked image due to space constraints in the manuscript. We have included a few more examples in our rebuttal PDF attached to the global response, and we would be happy to include these in an appendix in the camera-ready. Whether our autoencoder recovers semantically meaningful directions of variation in latent space is ultimately a subjective judgment, and we do not believe that readers would gain additional insight regarding our method's performance on this task by including quantitative metrics.
>
> *"I found the following two references that also compute nuclear norm without the SVD. Could the authors either mention these references in their manuscript or clarify why these aren’t relevant?"*
>
> We would be pleased to add these references to the related work section. However, note that both of these papers propose methods for nuclear norm regularization in matrix learning problems, where one seeks to learn a single matrix $A \in \mathbb{R}^{m\times n}$. In contrast, our work generalizes the efficient method of Rennie and Srebro [2005] to non-linear learning problems, where the appropriate analog to penalizing the nuclear norm of a matrix is penalizing the nuclear norm of the Jacobian of the function being learned.
>
> *"Figure 1 has only the color legend. Could the authors confirm (and explicitly state on their manuscript) that the x- and y-axes correspond to each coordinate of the input x?"*
>
> That is correct. We will clarify this in the camera-ready.
>
> *"The PSNR values in Table 1 are based on averaging over only 100 images. Is there a reason why more images were not used? Could the authors repeat the experiment by averaging over more images?"*
>
> The standard test sets in the denoising literature are quite small; for example, our other test set "CBSD68" contains 68 images. We built our Imagenet test set using 100 random images to approximately match test set sizes that are common in the denoising literature. We do not expect that the results of our comparison would be significantly different if we used a larger test set.
>
> We would be happy to incorporate the rest of your suggested changes to our paper's formatting in the camera-ready.
>
> We hope this answers your questions and would be pleased to continue this conversation in the author-reviewer discussion period.

---

> > ### Comment · Reviewer_Gitn · 2024-08-08
> >
> > Thank you for clearly addressing my comments!

---

### Official Review · Reviewer_P8Q4 · 2024-07-10

**Soundness:** 3
**Presentation:** 2
**Contribution:** 3
**Rating:** 7
**Confidence:** 4

**Summary:**

The paper proposes a computationally tractable method to induce low-rankness of a neural network's Jacobian. The method essentially generalizes the max-norm from Renni and Srebro, 2005, to more general compositions of functions, as it is common in neural networks. The method is made computationally efficient by estimating the Frobenius norm of the Jacobian. Experiments for denoising and representation learning are used to validate the efficacy of the method.

**Strengths:**

The idea of penalizing the rank of a function's Jacobian is practically useful in certain machine-learning applications. The proposed method enables one to do so in a computationally tractable manner that enables its use in applications where brute-force computation of an SVD and the Jacobian is infeasible.

The paper is technically solid and the method is evaluated on two realistic example tasks.

**Weaknesses:**

While the preliminaries in Section 3.1 are easy to follow, large parts of Section 3.2, which contains the main contribution, would be easier to follow if the authors would give a concrete example early on for the functions f and g. An illustrative example would make the idea more accessible.

One of the key arguments of the paper is that the method scales to high-dimensional problems and very large neural networks. The application examples, however, consider relatively small and simple problems with relatively simple architectures (for both the denoising of not-so-large images and autoencoder-based representation learning of images tasks). It is therefore not demonstrated that the method indeed scales to high-dimensional problems and complex neural networks.

The impact of estimating the Frobenius norm of the Jacobian (as detailed in Section 3.3) on the performance of applications was not studied. The authors claim that a single draw of eps is sufficient to evaluate (6), but it is unclear to me what the impact of this choice is.

It might be good if the authors could quantify the complexity of the proposed method, when used during training (either in run-time or in the number of operations/multiplications). This could also provide insight into the scalability to large-dimensional problems with very large neural networks.

**Questions:**

How are hyperparameters (e.g., the regularization factor eta) in the provided applications set?

Denoising and representation learning can be accomplished with other means as well, and existing state-of-the-art methods are likely to outperform the proposed approach. Is there any application that would uniquely benefit from the proposed regularizer, i.e., in which no other existing method can be used?

**Limitations:**

I do not see a specific limitation that was not discussed.

---

> ### Author Rebuttal · Authors · 2024-08-06
>
> Thank you for your thoughtful review of our manuscript. We are glad that you appreciate our technical contributions and the practical utility of our method.
>
> *"One of the key arguments of the paper is that the method scales to high-dimensional problems and very large neural networks. The application examples, however, consider relatively small and simple problems with relatively simple architectures (for both the denoising of not-so-large images and autoencoder-based representation learning of images tasks). It is therefore not demonstrated that the method indeed scales to high-dimensional problems and complex neural networks."*
>
> In our denoising experiment, we apply our regularizer to a function of the form $f_\theta : \mathbb{R}^d \rightarrow \mathbb{R}^d$, where $d = 3 \times 256 \times 256$ since it operates on $256 \times 256$ RGB images. In this case, $Jf_\theta[x]$ is a $(3 \times 256 \times 256) \times (3 \times 256 \times 256) = 196608 \times 196608$ matrix, which occupies over 154 GB of memory if stored in float32 format and therefore cannot be stored on most GPUs. As our regularizer does not require any Jacobian computations, it can be successfully applied to this problem. In contrast, naive Jacobian nuclear norm regularization would be intractable due to the need to compute and take the SVD of the $196608 \times 196608$ denoiser Jacobians.
>
> *"It might be good if the authors could quantify the complexity of the proposed method, when used during training (either in run-time or in the number of operations/multiplications). This could also provide insight into the scalability to large-dimensional problems with very large neural networks."*
>
> We provide the formula for our regularizer in Equation (6) in Section 3.3 of our manuscript. As $f(x) = g(h(x))$, one must evaluate the $h(x)$ and $g(h(x))$ terms to train $f$ regardless of the training objective. Our regularizer requires additionally computing $h(x+\epsilon)$ and $g(h(x) + \epsilon)$, so if one computes the regularizer with $N$ noise samples, our regularizer requires an additional $2N$ function evaluations. As we used $N=1$ noise sample to compute our regularizer in our denoising and autoencoding experiments, the marginal cost of our regularizer was 2 function evaluations per iteration.
>
> This compares favorably to computing the Jacobian, which requires $O(d)$ function evaluations for a function $f : \mathbb{R}^d \rightarrow \mathbb{R}^d$, and to the $O(d^3)$ SVD computation which is then required to compute the gradient of the Jacobian nuclear norm. Furthermore, as noted above, merely *storing* the Jacobian matrix is intractable for high-dimensional problems such as denoising.
>
> *"How are hyperparameters (e.g., the regularization factor eta) in the provided applications set?"*
>
> In the denoising experiments, we set $\eta = \sigma^2$, where $\sigma^2$ is the noise variance. (In our current manuscript, we have indicated that we set $\eta = \sigma$ in these experiments; this is a typo that we will correct in the camera-ready.) This choice of $\eta$ is motivated by results on optimal singular value shrinkage from Gavish and Donoho [2017], whose optimal shrinker solves a special case of our proposed denoising problem (11). (See our section "Singular value shrinkage" from line 230 onwards for details.) We set $\eta$ empirically in the autoencoder experiment via grid search.
>
> *"Denoising and representation learning can be accomplished with other means as well, and existing state-of-the-art methods are likely to outperform the proposed approach. Is there any application that would uniquely benefit from the proposed regularizer, i.e., in which no other existing method can be used?"*
>
> Our primary contribution is a tractable and well-grounded method for Jacobian nuclear norm regularization that can be applied to high-dimensional deep learning problems. There are few problems for which some given regularizer is strictly necessary and no other approach can be used, and our regularizer is no exception to this principle. Our goal in our experiments was to demonstrate the practical value of our proposed regularizer in high-dimensional learning problems by showing that it performs well in two tasks of interest to the machine learning community. The efficiency and simplicity of our method will enable the community to build on our work and discover new applications for Jacobian nuclear norm regularization.
>
> We hope this answers your questions and would be pleased to continue this conversation during the author-reviewer discussion period.

---

### Official Review · Reviewer_82sN · 2024-07-12

**Soundness:** 3
**Presentation:** 3
**Contribution:** 3
**Rating:** 6
**Confidence:** 3

**Summary:**

The authors present an efficient method for regularizing the Jacobian of deep networks such that it is low-rank. This work is motivated by the fact that penalizing the Jacobian by the nuclear norm regularization is in general a computationally difficult task, as it needs to (i) actually compute the Jacobian and (ii) take the SVD of a large matrix. The authors' proposed method comes from the observation that the nuclear norm of a matrix can be computed by a non-convex optimization problem, which I believe is somewhat commonly considered in the matrix factorization literature. Then, they propose their method for estimating the Jacobian Frobenius norm, which is equivalent to computing the nuclear norm (roughly speaking).

**Strengths:**

- I think the idea is neat and well-motivated. It stems from a tactic that I think is commonly used in matrix factorization literature (see [1] for example).
- The theoretical results seem to fit well with the main idea of the paper and overall strengthens the paper.
- I think this method could be a good starting point for future papers that need to consider computing the Jacobian (or at least regularize it). For example, there are papers in the topic of image editing, and I think there could be applications within that field that could use this method to circumvent costly evaluation of the Jacobian and computing its SVD.

[1] Lijun Ding, Dmitriy Drusvyatskiy, Maryam Fazel, Zaid Harchaoui. "Flat minima generalize for low-rank matrix recovery".

**Weaknesses:**

- I think the main weakness of the paper is its experimental section. While I think this method could be a good starting point for other future papers, the current experiments don't really sell the effectiveness of the method. It seems that one of the main benefits of this method as shown in the experimental results in Table 1 is that the proposed denoiser is almost as good as supervised denoiser, despite having no corresponding clean images for training. But isn't N2N also unsupervised in the sense that it doesn't need clean pairs of images? It doesn't seem to have impressive performance gains over N2N or BM3D.
- Building upon the previous weakness, I wonder if there is a way of showing the effectiveness of this method through other means -- for example computational efficiency. This paper was motivated by the fact that it can circumvent costly Jacobian + SVD computations. Could the authors show with a small scale experiment showing the computational gains? I think that would strengthen the paper.

**Questions:**

I do not have any specific questions besides the ones in the weaknesses.

**Limitations:**

The limitations are listed in the weaknesses section.

---

> ### Author Rebuttal · Authors · 2024-08-06
>
> Thank you for your thoughtful review of our manuscript. We are glad that you appreciate our theoretical contributions and view our method as well-motivated. In this rebuttal, we will address the questions in your review. If you are satisfied with our answers, we respectfully ask that you raise your score for our submission.
>
> *"It seems that one of the main benefits of this method as shown in the experimental results in Table 1 is that the proposed denoiser is almost as good as supervised denoiser, despite having no corresponding clean images for training. But isn't N2N also unsupervised in the sense that it doesn't need clean pairs of images? It doesn't seem to have impressive performance gains over N2N or BM3D."*
>
> As you note, our method performs nearly as well as a supervised denoiser and comparably to Noise2Noise (N2N), despite being trained exclusively on highly corrupted data without access to clean images. While N2N is also an unsupervised denoising method, training a denoiser using N2N requires *several noisy samples* of each clean image. Given a clean image $x$, these samples are of the form $x + \epsilon_i$, where the $\epsilon_i$ are distinct realizations of zero-mean noise. Such data is typically unavailable if one lacks access to the clean images, rendering N2N impractical for real-world applications. In contrast, our method requires only a *single* realization of each noisy image and can hence be applied to arbitrary datasets of noisy images, which can be easily obtained in the wild.
>
> *"This paper was motivated by the fact that it can circumvent costly Jacobian + SVD computations. Could the authors show with a small scale experiment showing the computational gains? I think that would strengthen the paper."*
>
> In our rebuttal PDF attached to the global response, we have included a figure comparing time per training step at batch size 1 for the denoising problem (11) using our regularizer and a naive Pytorch implementation of the Jacobian nuclear norm. We experiment with images drawn from Imagenet downsampled to sizes $S \times S$ for $S \in \{8,16,32,64\}$; our V100 GPU's memory overflows for $S \geq 128$. Whereas time per training step with the naive nuclear norm implementation rises to nearly 129 seconds per iteration for $64 \times 64$ images, each training step with our regularizer takes under 120 milliseconds. We additionally highlight that a key advantage of our regularizer is that it is tractable for problem sizes where simply computing the model Jacobian is infeasible. For example, in our denoising experiments, we trained our denoiser on $256 \times 256$ RGB images. The model Jacobian in this case is a $(3 \times 256 \times 256) \times (3 \times 256 \times 256) = 196608 \times 196608$ matrix, which occupies over 154 GB of memory if stored in float32 format and therefore cannot be stored on most GPUs. As our regularizer does not require any Jacobian computations, it can be applied to problems such as denoising where Jacobian computations are prohibitive.
>
> We hope this resolves your concerns. If you are satisfied with our response, we respectfully ask that you raise your score for our submission. Otherwise, we would be please to continue this discussion during the author-reviewer discussion period.

---

> > ### Comment · Reviewer_82sN · 2024-08-09
> >
> > Thank you for answering my questions along with the additional experiments; I have raised my score accordingly.

---

### Official Review · Reviewer_gXsR · 2024-07-12

**Soundness:** 2
**Presentation:** 4
**Contribution:** 4
**Rating:** 4
**Confidence:** 4

**Summary:**

The paper describes an elegant and very efficient numerical scheme for minimizing a regularization term taking the form of the nuclear norm of the Jacobian $\\|Jf [x]\\|_*$ of a function $f$ at an input $x$. The numerical scheme can be applied when the function is written as a composition of two functions $f=g\circ h$. The numerical scheme is supported by two theorems and experiments. The regularization term is supported by experiments.

**Strengths:**

The main body of the article is very well written and pleasant to read. The  regularization term is relevant and can be used in several contexts. The numerical scheme proposed by the authors is very effective. The experiments are convincing.

**Weaknesses:**

For me, the main weaknesses of the article lie in the proof of the theorems (see below).

Another point that is unclear to me is due to the discrepancy between Theorem 3.1 and its application. Specifically, as it is currently written, on the right-hand side of (4), there is an infimum on all smooth functions  $h$ ad $g$. The "latent space" (the output space of $h$ and the input space of $g$) is not specified, and my understanding is that the infimum is over the union of all possible "latent spaces". The infimum over this union can be much smaller than the infimum for a given "latent space"... and, in practice, only one latent space is considered. This point should be clarified. Perhaps the theorem can be adapted to allow the freedom to choose the latent space.

Similarly, in practice, neural networks (of fixed architecture) are not able to reach all the smooth functions. This could have an impact on the two infimums of (4). This limitation should at least be stated clearly.

Finally, throughout the article, the reader would welcome hints on the choice of $g$ and $h$. In neural networks, we have $h_L \\circ...\\circ h_2 \\circ h_1$ and the reader doesn't know whether the choice of $h$ and $g$ resumes to choosing a layer separating them? or whether the method is extended to more than one composition in some other way?

Robustness to adversarial attacks seems to be a natural advantage of the regularization term. The authors may wish to mention this, along with other perspectives they deem relevant, in the conclusion.

In (3) and all the developments, the authors write $l(f(x),x)$. It is much more common in machine learning to consider  $l(f(x),y)$, for an input/output pair $(x,y)$. The article would really gain to be extended to this setting.

In Theorem 3.2: I think it is a little o: $o(\\sigma^2)$, a quantity that goes to $0$ faster than $\\sigma^2$.

{\bf Comments on the appendices:}

Line 455: Please define AM-GM.

Line 457: Please remove 'compact'. It is the usual SVD.

Line 492: A sentence like 'Consider $z\\in\\Omega$' would be welcome.

In (14), last term: I think it is $f_m(z) + R_m^z(x)$, not  $f_m(x) + R_m^z(x)$. This leads to several similar changes which (I think) do not have serious consequences. The changes are in (16), twice in line 498, once in line 509, and once below line 514.

Line 503: Please replace 'These functions....' with 'The composition of these functions'.

Line 509: I think the calculation would be simpler if you replace $ \\|Jf_m[z_i] - Jf_m[x]\\|_{*} $

by the absolute value of the difference between $  \\| Jf_m[z_i]\\|_{*} $

and $ \\| Jf_m[x]\\|_{*}  $.

Calculus below Line 514: The layout is odd. Also, the calculation would gain clarity if you write the sum in $i$. and upper-bound each term.

Above Line 531 (and throughout the proof): Please write $\\epsilon \\rightarrow 0$ under the arrow in $\\rightarrow 0$.

Line 533: Please define RHS

Line 535: The fact that $\\|h^k_{m,\\epsilon} - h^k_m\\|_{L^1(\\Omega,\\mu)} \rightarrow 0$ is not clear to me. For instance, $\mu$ might involve a weighted Dirac mass at a point where $h^k_m$ is discontinuous. Although this paragraph is a sort of illustration, it is preferable to avoid saying something that might be false. Finally, at some other location in the proof, I think you need $\mu$ to be absolutely continuous.

Line 542: Please precise 'complement in $\\Omega$'.

Line 545: It is $g^k_{m,\\epsilon}$, not $g^k_{m,e}$.

Equation below Line 546: It is $B(h^k_m(x), \\epsilon)$ , not $B(h^k_m(x))$.

**A major issue:** From lines 543 to 549, the authors try to prove an inequality which, I think, cannot hold. The problematic intermediate step is in the inequality below line 546. It is

$\\|Jg^k_{m,\\epsilon} [H_m^k(x)]\\|_F \\leq \\sup_{y\\in h_m^k(\\Omega)  \\|Jg_m^k[y]\\|_F }.$

It cannot hold because $g^k_{m,\\epsilon}$ is a smooth approximation of $g^k_m$ which is typically discontinuous. It is possible to find $\epsilon$ close to $0$ and points $y$ such that $\\|Jg^k_{m,\\epsilon} [y]\\|_F$ is arbitrary large. To me, there is no guarantee that $h^k_m(x)$ avoids such points. Said differently, to me, the first inequality below Line 546 does not hold when $B(h^k_m(x),\\epsilon) \\cup V_j\\neq \\emptyset$, where $V_j$ is a Voronoi cell such that $h^k_m(x)\\not\\in V_j$, which your hypotheses do not exclude. By the way, you might want to provide the formula for the mollification and detail the calculations similar to the first inequality below Line 546.

Above line 550: It is $L^1(\\Omega,\\mu)$, not $L^1(\\Omega)$

Line 555: You state: 'As $h^k_{m,\\epsilon} \\rightarrow h^k_{m}$, $\\mu(\\Omega_1(m,k,\\epsilon)) \rightarrow 0$.'. It would be useful to state in which sense $h^k_{m,\\epsilon} \\rightarrow h^k_{m}$ and to prove the details of the arguments guaranteeing that $\\mu(\\Omega_1(m,k,\epsilon)) \\rightarrow 0$. As already said, I suspect you need $\\mu$ to be absolutely continuous for the conclusion to hold. If I am correct, this hypothesis should appear in Theorem 3.1.

**A major issue:** Line 568, you state $\\int_\\Omega \\|Jg^k_{m,\\epsilon} [h^k_{m,\\epsilon} (x)]  \\|^2_F d\\mu \\rightarrow \\int_\\Omega \\|Jg^k_{m} [h^k_{m} (x)]  \\|^2_F d\\mu$. The sense of the term on the right of the equality is not clear to me since $g^k_{m}$ is piecewise linear, and might even be discontinuous. Even worth, I think nothing excludes $h^k_m$ to be constant on one of the Voronoi cell $V$. If this happens and if the constant turns out to be a point such that $Jg^k_{m}$ does not exist, the problem occurs on $V$ and we generally have $\\mu(V) \\neq 0$. Concerning the term on the left of equality sign, I fear you encounter problems similar to those you may be familiar with since you mention the Total Variation in your article. The term on the left takes the jumps into account and might even go to infinity since you have a square. Looking at the proof of this statement, we find a possible cause for the mistake in lines 569-570. There, you state `...for any given $x\\in\\Omega$ one can choose $\\epsilon>0$ sufficiently small so that $g^k_{m,\\epsilon}(h^k_{m,\\epsilon} (x)) = g^k_m(h^k_{m} (x))$...'. This is not true if $x$ is on the boundary of the Voronoi cells defining $h^k_m$ or is such that $h^k_{m} (x)$ is the boundary of two Voronoi cells defining $g^k_m$.

Appendix A.3: You always use the (big) $O$ but sometimes you have to write the (little) $o$. You need to distinguish between the two notations.

**Questions:**

SVS and SVT are often interpreted as proximal operators. Can you please mention it in your introduction?

**Limitations:**

The main limitation is that there are gaps in the proof of the main theorem.

---

> ### Author Rebuttal · Authors · 2024-08-06
>
> Thank you for your thorough review of our paper. We would be pleased to make the edits you have suggested for clarity and fix the typos you have pointed out in the camera-ready version of our paper. We have updated our proof of Theorem 3.1 to address your major issues; as we cannot upload revised manuscripts during the rebuttal period, we summarize our revised arguments below.
>
> 1. Our proof requires $\mu$ to be absolutely continuous wrt the Lebesgue measure $\lambda$ to transfer results of the form $\lambda(E) = 0$ or $\lambda(E_n) \rightarrow 0$ for sets $E_n, E \subseteq \Omega$ to $\mu(E) = 0, \mu(E_n) \rightarrow 0$, resp. Thank you for pointing this out; we will add this hypothesis to Theorem 3.1 in the camera-ready.
>
> 2. For any Voronoi partition $V_i, i=1,...,N(k)$ and corresponding functions $h_m^k, g_m^k$ that we construct in our proof, the preimage of the union of Voronoi boundaries (which we call $S_m^k$) under $h_m^k$ is a set of Lebesgue measure zero. If $\mu \ll \lambda$, then the preimage has $\mu$-measure zero as well. Our reasoning is as follows:
>
> In lines 507-514, we define $g_m^k, h_m^k$ as piecewise affine functions such that $g_m^k(x) := g_m^{z_i}(x)$ and $h_m^k(x) := h_m^{z_i}(x)$ for all $x \in \textrm{int}(V_i)$. The affine functions $g_m^{z_i}, h_m^{z_i}$ have the key property that $\frac{\eta}{2}\left( \|Jg_m^{z_i}[h_m^{z_i}(x)]\|_F^2 + \|Jh_m^{z_i}[x]\|_F^2 \right) = \eta \|J f_m[z_i]\|\_*.$
>
> At any $x$ on the interior of a Voronoi cell $V_i$, $g_m^k$ and $h_m^k$ are equal to $g_m^{z_i}, h_m^{z_i}$, resp, so by the key property above, $\frac{\eta}{2}\left( \|Jg_m^k[h_m^k(x)]\|_F^2 + \|Jh_m^k[x]\|_F^2 \right) = \eta \|J f_m[z_i]\|\_* < + \infty$. In particular, $\|Jg_m^k[h_m^k(x)]\|_F^2$ and $\|Jh_m^k[x]\|_F^2$ must both be well-defined and finite at this $x$. This can only happen if $h_m^k(x)$ lies on the interior of a Voronoi cell, as otherwise $Jg_m^k[h_m^k(x)]$ would be undefined. Hence $h_m^k$ maps the interiors of Voronoi cells to the interiors of Voronoi cells; the contrapositive is that if $h_m^k(x)$ lies on a Voronoi boundary, then $x$ also lies on a Voronoi boundary. Consequently, the preimage of the Voronoi boundaries under $h_m^k$ is a set of Lebesgue measure zero. If $\mu \ll \lambda$, this is also a set of $\mu$-measure zero.
>
> Using this fact, we can replace all integrals over $\Omega$ from line 525 onwards with integrals over $\Omega(m,k) := \Omega \setminus S_m^k$ without changing their value. In particular, $Jg_m^k[h_m^k(x)]$ is well-defined $\mu$-ae, which should resolve your concerns re: line 568, for which you note *"I think nothing excludes $h_m^k$ to be constant on one of the Voronoi cell $V$. If this happens and if the constant turns out to be a point such that $Jg_m^k$ does not exist, the problem occurs on $V$ and we generally have $\mu(V) = 0$."* It should also resolve your concerns re: lines 569-570, for which you state that one cannot always find $\epsilon>0$ so that $g_{m,\epsilon}^k[h_{m,\epsilon}^k(x)] = g_m^k[h_m^k(x)]$. One can in fact find such $\epsilon$ for $\mu$-almost all $x$, which is sufficient to apply the dominated convergence theorem.
>
> 3. We use an alternative approach to show that $\|g_{m,\epsilon}^k \circ h_{m,\epsilon}^k - g_{m,\epsilon}^k \circ h_{m}^k\|_{L^1(\Omega,\mu)} \rightarrow 0$ that avoids the issue you highlight re: lines 543-549. We employ a different decomposition of $\Omega \setminus S_m^k$ into good and bad sets:
>
> - The good set $\Omega_0(m,k,\epsilon) \subseteq \Omega(m,k)$ is the set of $x \in \Omega(m,k)$ such that $d(x, S_m^k) > \epsilon$. ($d(p,S)$ denotes the distance from point $p$ from set $S$.)
> - The bad set $\Omega_1(m,k,\epsilon)$ is $\Omega(m,k) \setminus \Omega_0(m,k,\epsilon) = \(x \in \Omega(m,k) : d(x, S_m^k) \leq \epsilon \)$.
>
> For all $x \in \Omega_0(m,k,\epsilon)$, $h_{m,\epsilon}^k(x) = h_m^k(x)$ because $d(x, S_m^k) > \epsilon$ and we employ the standard mollifier supported on $B(0,\epsilon)$. Consequently, $g_{m,\epsilon}^k(h_{m,\epsilon}^k(x)) = g_{m,\epsilon}^k(h_m^k(x))$ and therefore $\|g_{m,\epsilon}^k(h_{m,\epsilon}^k(x)) - g_{m,\epsilon}^k(h_{m}^k(x)) \|\_2 = 0$, so $\int_{\Omega_0(m,k,\epsilon)} \|g_{m,\epsilon}^k(h_{m,\epsilon}^k(x)) - g_{m,\epsilon}^k(h_{m}^k(x)) \|_2 d\mu = 0$. We no longer need the reasoning in lines 543-549 to prove this integral converges to 0.
>
> We now address $\int_{\Omega_1(m,k,\epsilon)} \|g_{m,\epsilon}^k(h_{m,\epsilon}^k(x)) - g_{m,\epsilon}^k(h_{m}^k(x)) \|\_2 d\mu$. We employ the same bound on the integrand as under line 552 for the new bad set $\Omega_1(m,k,\epsilon)$. To see that $\mu(\Omega_1(m,k,\epsilon)) \rightarrow 0$, note that $\lambda(\Omega_1(m,k,\epsilon)) \rightarrow 0$, as $\Omega_1(m,k,\epsilon)$ is a union of cylinders of radius $\epsilon$ centered at the Voronoi boundaries $S_m^k$, which have Lebesgue measure zero. Under our new assumption $\mu \ll \lambda$, it follows that $\mu(\Omega_1(m,k,\epsilon)) \rightarrow 0$.
>
> These results jointly show that $\|g_{m,\epsilon}^k \circ h_{m,\epsilon}^k - g_{m,\epsilon}^k \circ h_{m}^k\|_{L^1(\Omega,\mu)}$ while avoiding the issue you highlight re: lines 543-549.
>
> We hope this resolves your concerns regarding our proof. If you are satisfied with our response, we respectfully ask that you raise your score for our submission. Otherwise, we would be please to continue this discussion during the author-reviewer discussion period.

---

> > ### Comment · Reviewer_gXsR · 2024-08-11
> >
> > I have read the author's rebuttal but, in my opinion, the proof needs to be re-read in detail and the article needs another round of review. It is not possible, from an article and a rebuttal, to check that the proof is correct. For this reason, I will not change my rating. To avoid errors in the futur, I recommend writing down the details of the mollification and being careful when swapping limits and integrals.
> >
> > Best regards,

---

> > > ### Author Response · Authors · 2024-08-12
> > >
> > > We thank Reviewer gXsR for reading our rebuttal. We are confident that our updated proof is now correct, and we believe the details we have provided in our rebuttal are sufficient to resolve the specific concerns raised by Reviewer gXsR under the "major issue" headings.

---

### Author Rebuttal · Authors · 2024-08-06

Thank you for your thoughtful reviews of our submission. We have attached a PDF containing four figures:

1. For Reviewer 82sN, we have included a figure (top-left) comparing time per training step at batch size 1 for the denoising problem (11) using our regularizer and a naive Pytorch implementation of the Jacobian nuclear norm. We experiment with images drawn from Imagenet downsampled to sizes $S \times S$ for $S \in \{8,16,32,64\}$; our V100 GPU's memory overflows for $S \geq 128$. Whereas time per training step with the naive nuclear norm implementation rises to nearly 129 seconds per iteration for $64 \times 64$ images, each training step with our regularizer takes under 120 milliseconds.

2. For Reviewer Gitn, we have included three more sets of latent traversals in our autoencoder's latent space (top-right, bottom-left, and bottom-right).

---

### Decision · Program_Chairs · 2024-09-25

**Decision:**

Accept (poster)

**Comment:**

This paper addresses the computational challenges in regularizing deep learning with the nuclear norm of the Jacobian matrix.  The key discovery is that that for functions parametrized as compositions $f = g \circ h$, the regularizer is equivalent to the average squared Frobenius norms of $Jg$ and $Jh$.  The latter two are then efficiently computed via a denoising-style approximation.  Empirical superiority is also shown in denoising and representation learning.

The paper addresses a common challenge with a neat and effective solution.  The new proof in the rebuttal to Reviewer gXsR appears correct in my opinion, and I recommend the authors to rewrite the paper to reflect the changes.  Although the experiments can still be much improved as the reviewers suggested, it appears agreed that the merits of the paper are sufficient to warrant publication at the conference.